# Inflammation and cytomegalovirus viremia during pregnancy drive sex-differentiated differences in mortality and immune development in HIV-exposed infants

Children who are HIV-exposed but uninfected have increased infectious mortality compared to HIV-unexposed children, raising the possibility of immune abnormalities following exposure to maternal viraemia, immune dysfunction, and co-infections during pregnancy. In a secondary analysis of the SHINE trial in rural Zimbabwe we explored biological pathways underlying infant mortality, and maternal factors shaping immune development in HIV-exposed uninfected infants. Maternal inflammation and cytomegalovirus viraemia were independently associated with infant deaths: mortality doubled for each $\log_{10}$ rise in maternal C-reactive protein (adjusted hazard ratio (aHR) 2.09; 95% CI 1.33–3.27), and increased 1.6-fold for each $\log_{10}$ rise in maternal cytomegalovirus viral load (aHR 1.62; 95% CI 1.11–2.36). In girls, mortality was more strongly associated with maternal C-reactive protein than cytomegalovirus; in boys, mortality was more strongly associated with cytomegalovirus than C-reactive protein. At age one month, HIV-exposed uninfected infants had a distinct immune milieu, characterised by raised soluble CD14 and an altered CD8 + T-cell compartment. Alterations in immunophenotype and systemic inflammation were generally greater in boys than girls. Collectively, these findings show how the pregnancy immune environment in women with HIV underlies mortality and immune development in their offspring in a sex-differentiated manner, and highlights potential new intervention strategies to transform outcomes of HIV-exposed children. ClinicalTrials.gov/ NCT01824940.

HIV has been a major contributor to infant mortality in sub-Saharan Africa for several decades. Although vertical transmission has not been eliminated, the scale-up of maternal antiretroviral therapy (ART) means over 15 million children are now HIV-exposed but uninfected (HEU)[1]. There is growing evidence that HEU infants have disparities in health outcomes compared to HIV-unexposed infants[2–6], including higher mortality from common childhood infections and an unusual spectrum of pathogens[2], raising the possibility of immune abnormalities[7].

We recently compared clinical outcomes between infants who were HIV-exposed and HIV-unexposed in the Sanitation Hygiene Infant Nutrition Efficacy (SHINE) trial in rural Zimbabwe. Despite high coverage of maternal ART and uptake of exclusive breastfeeding, mortality in infants exposed to HIV was 41% higher than in infants not exposed to HIV[8]. Infants who survived and remained HIV-free had impaired growth and development[8,9]. Whether the ongoing associations between maternal HIV and infant outcomes in the current ART

✉e-mail: ceri.evans@qmul.ac.uk

era are driven through HIV-related biological pathways or are reflective of sociodemographic differences remains uncertain[2,3].

Here, in a large cohort of infants in rural Zimbabwe, we identify specific biological mechanisms that leave an immune 'footprint' on the HIV-exposed infant, opening up potential new intervention targets to transform outcomes of HIV-exposed infants.

## Results

### Maternal systemic inflammation is associated with infant mortality in HIV-exposed but not in HIV-unexposed infants

Despite maternal ART coverage of over 80% in the SHINE trial, which is representative of national ART coverage in Zimbabwe[1], we previously reported that the risk of mortality was 41% higher among HIV-exposed compared to HIV-unexposed infants[8]. Among 726 pregnant women with HIV (Supplementary Table 1), 579 (80%) had samples available for HIV viral load testing at a median gestational age of 16 weeks (interquartile range (IQR) 13, 21); 322/579 (56%) had detectable HIV viremia above the assay limit of detection (150 copies/mL). For each log rise in maternal HIV viral load, the risk of mortality in infants increased by 41% (adjusted hazard ratio (aHR) 1.41; 95% confidence interval (CI) 1.02, 1.94; $P = 0.04$). Infants born to mothers with detectable vs. undetectable HIV viremia had twice the risk of mortality (8.2% vs. 4.2%; aHR 1.95; 95% CI 0.82, 4.63; $P = 0.13$) (Table 1 and Fig. 1A, B). However, the population-attributable fraction of mortality due to HIV viremia was 38.8%, suggesting that even if all pregnant women were receiving ART and had viral suppression, excess infant mortality would remain. We therefore explored alternative potential mechanisms underlying mortality.

Immunodeficiency and inflammation are distinct pathways leading to mortality in children[10] and adults[11] with HIV infection, and inflammation frequently persists despite ART[12,13]. We therefore hypothesised that maternal inflammation during pregnancy is associated with infant mortality. We measured C-reactive protein (CRP), which is produced by the liver in response to inflammatory cytokines such as IL-6[14], and soluble CD14 (sCD14), which is a glycoprotein shed from activated monocytes and macrophages[15]. Women with HIV had more systemic inflammation during the second trimester of pregnancy than women without HIV (median plasma CRP 4.72 mg/L (IQR 1.54, 11.85) vs. 2.67 mg/L (1.04, 6.02), $P < 0.001$; median sCD14 1.36 mg/L (1.09, 1.69) vs. 1.06 (0.87, 1.32), $P < 0.001$) (Fig. 1C, D). Among 621 HIV-exposed infants, maternal CRP during pregnancy was independently associated with mortality through 18 months of age. After adjusting for maternal HIV viral load, CD4 count and CMV co-infection, the risk of infant mortality doubled for each log rise in maternal CRP (aHR 2.09; 95% CI 1.33, 3.27). When infants were dichotomised into those born to mothers with CRP concentrations above or below the median value, a high compared to a low CRP was strongly associated with mortality (aHR 3.53; 95% CI 1.73,

7.19). However, there was no association between maternal plasma sCD14 and the risk of infant mortality (Table 1 and Fig. 1E, F). In contrast to HIV-affected mother-infant pairs, there was no association between maternal CRP and mortality in infants born to mothers without HIV (Table 2), suggesting an HIV-specific association between maternal inflammation and infant mortality.

### HIV-exposed uninfected infants have a distinct inflammatory milieu

Given the marked associations between maternal inflammation and infant mortality, we hypothesised that the altered antenatal immune milieu among women with HIV affects the developing fetal immune system, putting HEU infants at risk of death from infection in early life. We first focused on defining the infant systemic inflammatory milieu using plasma samples collected at 1 month of age. sCD14 concentrations were significantly higher in HEU compared to HIV-unexposed infants (mean 1.10 mg/L (standard deviation (SD) 0.4) vs. 0.85 mg/L (SD 0.4), adjusted difference 0.25 mg/L, 95% CI 0.15, 0.35; $P < 0.001$; Fig. 2A) at one month of age. Concentrations of soluble CD163 (sCD163), another glycoprotein associated with monocyte and macrophage activation[16], also tended to be higher in HEU compared to HIV-unexposed infants, but absolute differences were small and statistical evidence was weaker (mean 781.7 pg/mL (SD 343) vs. 721.2 pg/mL (SD 286), adjusted difference 67.35 pg/mL, 95% CI −8.56, 143.25; $P = 0.08$, two-sided; Fig. 2B). There were weak correlations between maternal and infant sCD14 (Spearman's *rho* 0.15, $P = 0.09$, two-sided) and maternal and infant sCD163 (Spearman's *rho* 0.21, $P = 0.02$, two-sided). In contrast, plasma IL-6 (a pro-inflammatory cytokine which stimulates production of CRP) did not significantly differ between HEU and HIV-unexposed infants ($P = 0.65$, two-sided), and CRP concentrations were lower in HEU infants compared to HIV-unexposed infants (2.44 mg/L (SD 7.6) vs. 4.10 mg/L (SD 13.0); adjusted Tobit β 0.53, 95% CI 0.29, 0.96; $P = 0.37$). There were no correlations between maternal and infant IL-6 (Spearman's *rho* 0.10, $P = 0.25$) or CRP concentrations (Spearman's *rho* −0.03, $P = 0.75$).

Taken together, the systemic inflammatory milieu in HIV-exposed infants was distinct from that of HIV-unexposed infants, with significantly higher sCD14, and lower CRP concentrations. We reasoned that co-trimoxazole exposure during pregnancy could be one explanation for lower CRP concentrations since we have previously shown that co-trimoxazole lowers CRP through its combined antimicrobial and anti-inflammatory properties[17].

### Perinatal HIV exposure is associated with monocyte activation and a distinct pattern of enteropathy biomarkers

Since the systemic inflammatory milieu of HEU infants is characterised by raised plasma concentrations of the innate pathogen-recognition

**Table 1 | Associations between maternal HIV biomarkers in the second trimester and infant mortality through 18 months**

| Maternal biomarker | Child mortality *n/N* (%) | Hazard ratio | *P* | Adjusted hazard ratio | *P* |
|---|---|---|---|---|---|
| HIV viral load undetectable | 11/260 (4.2%) | Reference | | Reference | |
| Detectable ≥150 copies/mL | 27/330 (8.2%) | 1.96 (0.89, 4.34) | 0.097 | 1.95 (0.82, 4.63)[a] | 0.130 |
| C-reactive protein <4.72 mg/L | 9/318 (2.8%) | Reference | | Reference | |
| C-reactive protein ≥4.72 mg/L | 32/318 (10.1%) | 3.64 (1.86, 7.13) | <0.001 | 3.53 (1.73, 7.19)[b] | 0.001 |
| Soluble CD14 < 1.36 mg/L | 20/318 (6.3%) | Reference | | Reference | |
| Soluble CD14 ≥ 1.36 mg/L | 21/318 (6.6%) | 1.06 (0.56, 2.04) | 0.851 | 1.04 (0.53, 2.05)[c] | 0.914 |
| Log rise in HIV viral load | N/A | 1.40 (1.05, 1.86) | 0.022 | 1.41 (1.02, 1.94)[a] | 0.035 |
| Log rise in C-reactive protein | N/A | 2.31 (1.52, 3.52) | <0.001 | 2.09 (1.33, 3.27)[b] | 0.001 |
| Log rise in soluble CD14 | N/A | 1.65 (0.25, 10.75) | 0.599 | 1.65 (0.25, 10.80)[c] | 0.600 |

Cox proportional hazard models were used to evaluate associations. Presented *P*-values have not been adjusted for multiple comparisons.
[a]Adjusted for maternal age and education, household socioeconomic status, and gestational age at blood sampling and randomised trial arm.
[b]Adjusted for maternal HIV viral load, CD4 count, CMV co-infection, gestational age at blood sampling and randomised trial arm.
[c]Adjusted for maternal HIV viral load, CD4 count, gestational age at blood sampling and randomised trial arm.

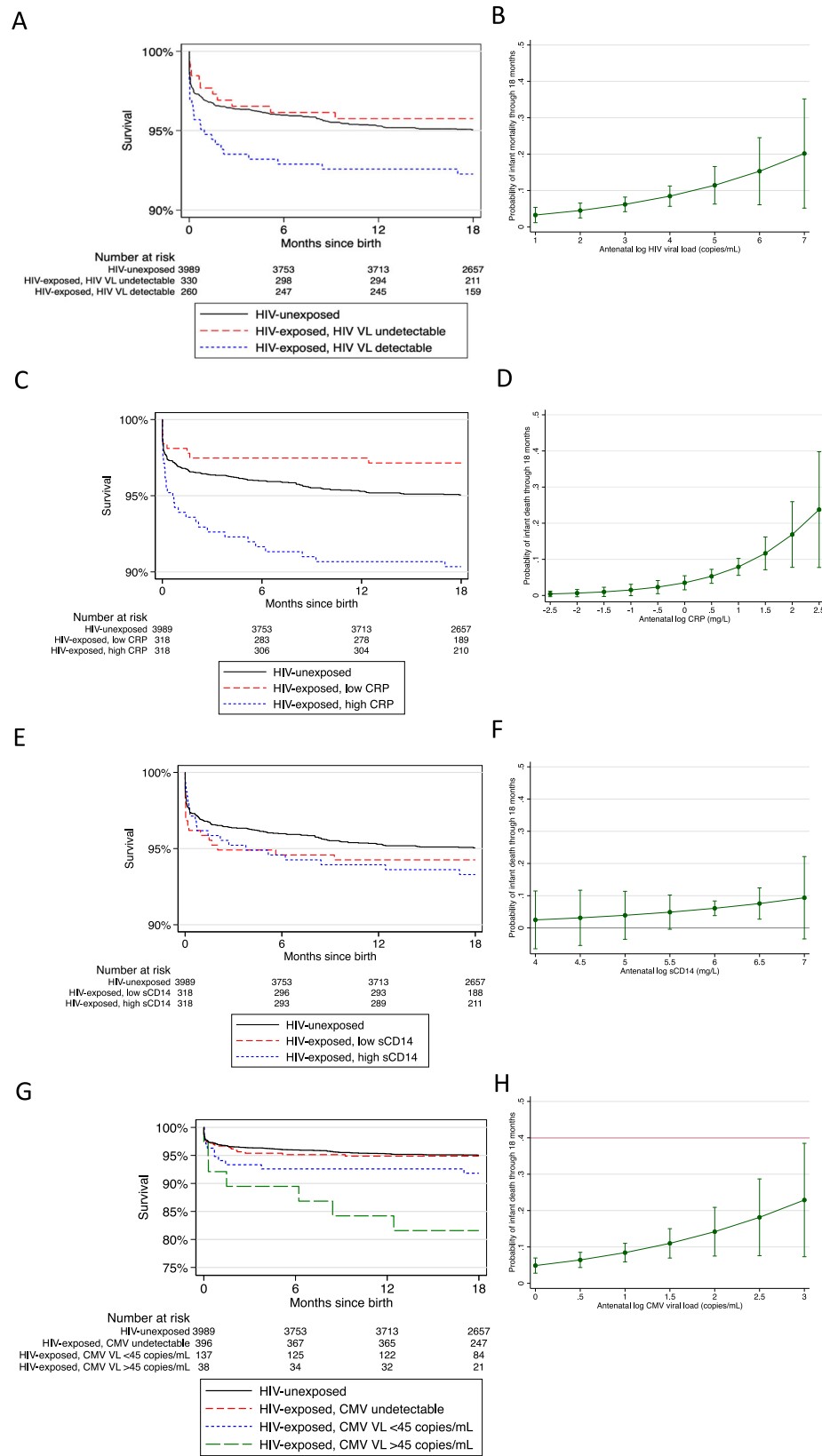

receptors/co-receptors sCD14 and sCD163, we next explored cellular markers of monocyte activation. Cryopreserved peripheral blood cells for 101 HEU infants and 90 HIV-unexposed infants were evaluated by flow cytometry, at a median age of 39 (IQR 35, 56) days and 39 (IQR 35, 58) days, respectively. There are three sub-populations of blood monocytes that reflect a developmental continuum from least

to most differentiated (classical, intermediate, non-classical monocytes); more differentiated monocyte subsets, particularly the intermediate phenotype, are expanded in inflammatory[18–20] diseases. There were minimal differences in monocyte subsets between HEU and HIV-unexposed infants; compared to monocyte populations in HIV-unexposed infants, the classical monocyte population in HEU infants

**Fig. 1 | Associations between maternal biomarkers and infant mortality.**
**A** Kaplan–Meier estimates for the association between maternal HIV viremia in the second trimester and infant mortality through 18 months. HIV-unexposed children are shown with black line for reference. Red: HIV-exposed, maternal HIV viral load undetectable; blue: HIV-exposed, maternal HIV viral load detectable. **B** Predictive probability of child mortality with increasing antenatal HIV viral load in the second trimester. Graph shows point estimates with 95% error bars. **C** Kaplan–Meier estimates for the association between maternal C-reactive protein in the second trimester and infant mortality through 18 months. HIV-unexposed children are shown with a black line for reference. Red: HIV-exposed, maternal C-reactive protein below median value; blue: HIV-exposed, maternal C-reactive protein above median value. **D** Predictive probability of child mortality with increasing antenatal C-reactive protein in the second trimester. Graph shows point estimates with 95% error bars. **E** Kaplan–Meier estimates for the association between maternal soluble CD14 in the second trimester and infant mortality through 18 months. HIV-unexposed children are shown with a black line for reference. Red: HIV-exposed, maternal soluble CD14 below median value; blue: HIV-exposed, maternal soluble CD14 above median value. **F** Predictive probability of child mortality with increasing antenatal soluble CD14 in the second trimester. The graph shows point estimates with 95% error bars. **G** Kaplan–Meier estimates for the association between maternal CMV viral load in the second trimester and infant mortality through 18 months. HIV-unexposed children are shown with a black line for reference. Red: HIV-exposed, maternal CMV viral load undetectable; blue: HIV-exposed, maternal CMV viral load <45 copies/mL; green: CMV viral load ≥45 copies/mL. **H** Predictive probability of child mortality with increasing antenatal CMV viral load in the second trimester. The graph shows point estimates with 95% error bars, CMV: cytomegalovirus; CRP: C-reactive protein; sCD14: soluble CD14; VL: viral load. Source data are provided as a Source Data file.

**Table 2 | Association between maternal biomarkers and mortality in children born to mothers with and without HIV**

| Maternal biomarker | HIV exposure | Concentration (median, IQR), mg/L | | Hazard ratio (95% CI) per log rise | *P* |
|---|---|---|---|---|---|
| | | **Died** | **Survived** | | |
| C-reactive protein | HIV-unexposed | 3.25 (1.00, 6.32) | 2.63 (1.04, 5.95) | 1.04 (0.75, 1.44) | 0.811 |
| | HIV-exposed | 9.18 (5.28, 18.02) | 4.35 (1.50, 10.99) | 2.31 (1.52, 3.52) | <0.001 |
| Soluble CD14 | HIV-unexposed | 1.09 (0.92, 1.38) | 1.06 (0.87, 1.32) | 1.94 (0.44, 8.57) | 0.382 |
| | HIV-exposed | 1.36 (1.11, 1.89) | 1.36 (1.09, 1.68) | 1.65 (0.25, 10.75) | 0.599 |

Cox proportional hazards models were used to evaluate associations. Presented *P*-values have not been adjusted for multiple comparisons.

was 4.4 percentage points lower (95% CI 0.2, 8.7), although the evidence for this was weak; *P* = 0.04 (Fig. 2L). Proportions of intermediate and non-classical monocytes were correspondingly higher in HEU infants, but neither was individually significantly higher compared to HIV-unexposed infants (Fig. 2M, N).

Median fluorescent intensities (MFI) of cell surface-expressed monocyte activation markers (CD40, CD86 and HLA-DR) on each monocyte subset were also compared between HEU and HIV-unexposed infants. There were no significant differences in MFI of CD40, CD86 or HLA-DR between HEU infants and HIV-unexposed infants in classical, intermediate or non-classical monocytes (Supplementary Table 2).

We next explored whether monocyte activation in HEU infants at the 1-month study timepoint is related to gut damage and subsequent transfer of LPS into the circulation due to loss of intestinal integrity[21]. We have previously shown that an inflammatory syndrome of the gut, termed environmental enteric dysfunction (EED), is almost ubiquitous among HIV-unexposed infants in this cohort[22]. Since HIV also causes a profound enteropathy, we hypothesised that gut health may be particularly impacted in infants exposed to HIV during pregnancy. We therefore evaluated a range of EED biomarkers in 145 HEU infants and 664 HIV-unexposed infants with available stool samples, and 102 HEU and 483 HIV-unexposed infants with available plasma samples (total 156 and 720 infants included in the EED sub-study). We identified a distinct pattern of enteropathy biomarkers in HEU compared to HIV-unexposed infants, characterised by more intestinal inflammation (raised faecal neopterin; Tobit β 1.39; 95% CI 1.14, 1.69, *P* = 0.001), differences in small intestinal villous damage (lower stool I-FABP; Tobit β 0.79; 95% CI 0.72, 0.87, *P* < 0.001) and some evidence of heightened epithelial regeneration (raised faecal REG-1β; Tobit β 1.47; 95% CI 0.95, 2.28, *P* = 0.088). We found no evidence of a difference in faecal myeloperoxidase (Tobit β 1.07; 95% CI 0.71, 1.61, *P* = 0.746) or faecal alpha-1 anti-trypsin (Tobit β 1.11; 95% CI 0.81, 1.52 *P* = 0.519).

Overall, despite the striking elevations in systemic sCD14 in early infancy, we found only small differences in monocyte subsets and phenotype between HEU and HIV-unexposed infants. HIV exposure is associated with a distinct pattern of enteropathy biomarkers, but we did not find evidence that enteropathy underlies the altered monocyte profile of HEU infants.

## Perinatal HIV exposure is associated with an altered CD8⁺ but not CD4⁺ T-cell compartment

Since HIV leads to profound T-cell activation, differentiation, and exhaustion in adults[23–25] and children[26], we next investigated associations between perinatal HIV exposure and the CD4 and CD8 T-cell compartments in their uninfected infants.

Cryopreserved peripheral blood cells for 117 HEU infants and 102 HIV-unexposed infants were evaluated by flow cytometry, at median ages of 38 (IQR 35, 54) days and 40 (IQR 35, 58) days, respectively. A higher proportion of CD8⁺ T-cells in HEU compared to HIV-unexposed infants expressed activation markers (CD38⁺/HLA-DR⁺ Fig. 2C) or proliferation markers (Ki67⁺, Fig. 2D). In HEU infants, 12.2% of CD8⁺ T-cells co-expressed CD38 and HLA-DR, which was 6.3 percentage points higher than HIV-unexposed infants (6.5%; *P* = 0.001). In HEU infants, 15.2% of CD8+T-cells expressed Ki67, compared to 9.3% in HIV-unexposed infants (*P* = 0.004). Compared to CD8⁺ T-cell populations in HIV-unexposed infants, a lower proportion of CD8⁺ T-cells in HEU infants were naïve (CD45RA⁺/CD27⁺), and higher proportions were central memory cells (CD45RA⁻/CD27⁺), effector memory cells (CD45RA⁻/CD27⁻), or terminally differentiated effector memory (TEMRA) cells (CD45RA⁺/CD27⁻) (Fig. 2E–H). Compared to CD8⁺ T-cell populations in HIV-unexposed infants, a smaller proportion was non-senescent (CD28⁺/CD57⁻, *P* = 0.008) (Fig. 2I–K). In contrast to CD8⁺ T-cells, there were no differences in CD4⁺ T-cell immunophenotype between infants who were HEU and those who were HIV-unexposed (Fig. 2O–W).

To further interrogate the role of maternal HIV disease severity in shaping the infant CD8⁺ T-cell compartment, we explored associations between maternal and infant biomarkers. There were no associations between second-trimester maternal HIV viral load and infant CD8⁺ T-cell subsets (Supplementary Table 3), but there was an association between third-trimester HIV viral load and CD8⁺ T-cell senescence (Supplementary Table 3). Finally, associations were evaluated between maternal immune activation and infant immune activation. Among HIV-affected mother-infant pairs, there were small-to-moderate associations between maternal and infant CD8⁺ T-cell immunophenotypes. The strongest associations were for proportions of Ki67⁺ (Spearman's *rho* 0.29, *P* = 0.008) and CD28⁻/CD57⁺ cells (Spearman's *rho* 0.31, *P* = 0.003) (Supplementary Table 4). By contrast, there were no

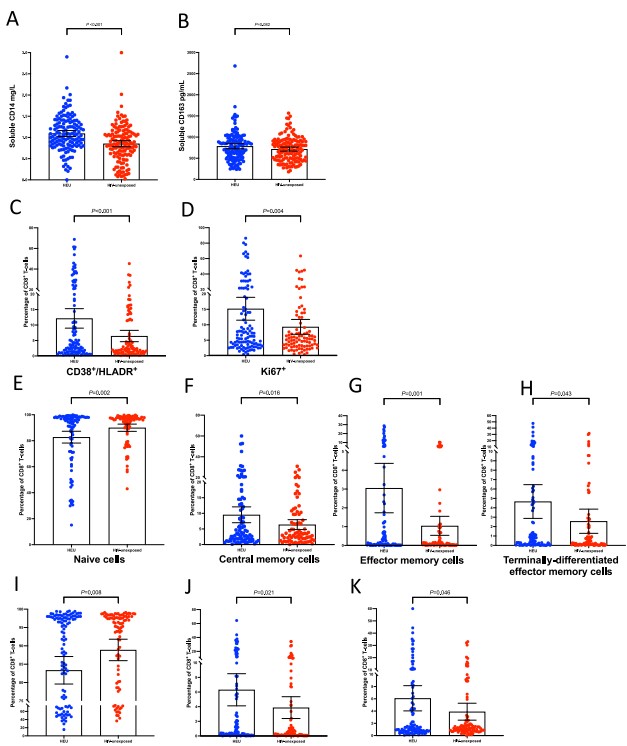

**Fig. 2 | Monocyte and T-cell activation in HEU and HIV-unexposed children at 1 month of age.** Bars at mean with 95% confidence intervals. Compared using fractional regression. *P*-values adjusted for infant sex, exact age at time of blood collection and randomised trial arm. All statistical tests are two-sided, and presented *P*-values have not been adjusted for multiple comparisons. **A** comparison of soluble CD14 between HEU and HIV-unexposed children; **B** comparison of soluble CD163 between HEU and HIV-unexposed children; **C** comparison of CD8[+] T-cell activation between HEU and HIV-unexposed children; **D** comparison of CD8[+] T-cell proliferation between HEU and HIV-unexposed children; **E–H** comparison of CD8[+] T-cell differentiation between HEU and HIV-unexposed children; **I, J** comparison of CD8[+] T-cell senescence between HEU and HIV-unexposed children; **K** comparison of CD8[+] T-cell exhaustion between HEU and HIV-unexposed children; **L–N** comparison of monocyte subsets between HEU and HIV-unexposed children; **O** comparison of CD4[+] T-cell activation between HEU and HIV-unexposed children; **P** comparison of CD4[+] T-cell proliferation between HEU and HIV-unexposed children; **Q–T** comparison of CD4[+] T-cell differentiation between HEU and HIV-unexposed children; **U, V** comparison of CD4[+] T-cell senescence between HEU and HIV-unexposed children; **W** comparison of CD4[+] T-cell exhaustion between HEU and HIV-unexposed children. Classical monocytes: CD3[−]/CD19[−]/CD20[−]/CD56[−]/HL-DR[+]/CD14[++]/CD16[−]; intermediate monocytes: CD3[−]/CD19[−]/CD20[−]/CD56[−]/HLADR[+]/CD14[++]/CD16[+]; non-classical monocytes: CD3[−]/CD19[−]/CD20[−]/CD56[−]/HLADR[+]/CD14[+]/CD16[++]. Non-senescent: CD28+/CD57-; senescent: CD28-/CD57+; exhausted: PD-1[+]. Activated: CD38[+]/HLA-DR[+]; proliferating: Ki67[+]. HEU: HIV-exposed uninfected. Blue: HEU children; red: HIV-unexposed children. Source data are provided as a Source Data file.

associations between maternal and infant CD8[+] T-cell or monocyte immunophenotypes in HIV-unexposed mother-infant pairs (Supplementary Table 4). Taken together, these findings therefore suggest that the skewed maternal immune milieu secondary to HIV shapes early infant immune development. This could be directly attributable to HIV or indirectly, for example, due to co-infections or the bystander immune activation that characterises HIV infection.

## Maternal CMV viremia in pregnancy is associated with infant mortality

To explore the effect of maternal co-infections on HEU infant outcomes, we focused on cytomegalovirus (CMV). CMV co-infection is a co-factor in HIV disease progression and is strongly associated with clinical outcomes in people with HIV, largely through effects on immune activation[27,28]. CMV seroprevalence in adulthood is close to 100% in sub-Saharan Africa[29], and CMV frequently reactivates in pregnancy. Among the whole cohort of 726 women with HIV, 563 (77%) had a plasma sample available for CMV DNA PCR at a median gestation of 16 weeks (IQR 13, 21). Among these 563 pregnant women, 174 (31%) had detectable CMV viraemia. By contrast, CMV viraemia was much less common in HIV-negative women in pregnancy (19/136 (14%); *P* < 0.001).

Among HIV-positive women with detectable CMV viraemia, 137/174 (79%) had a CMV viral load <45 copies/mL (the assay limit of

quantification), and 37/174 (18%) had a CMV viral load ≥45 copies/mL. For each log rise in maternal CMV viral load, infant mortality increased by 62% after adjusting for maternal HIV viral load, CD4 count, gestational age at blood sampling and randomised trial arm (aHR 1.62, 95% CI 1.11, 2.36; *P* = 0.01). When CMV viremia was dichotomised (detectable vs. undetectable viremia), the presence of maternal CMV was associated with 2-fold higher infant mortality (aHR 2.11, 95% CI 1.09, 4.09; *P* = 0.03). When CMV viremia was trichotomised (undetectable, detectable <45 copies/mL, detectable ≥45 copies/mL), there was a dose-response relationship with mortality (Table 3). When compared with infants born to mothers without HIV, CMV co-infection had a striking association with the risk of infant mortality (Fig. 1G). Finally, we assessed CMV later in pregnancy: 364 women with HIV had plasma CMV measured in the third trimester, with a similar proportion found to be CMV viremic as in the second trimester (96/358; 27%), compared to 14/85 (16%) amongst women without HIV (*P* = 0.04). CMV viremia in the third trimester among women with HIV was also associated with infant mortality (aHR 3.40, 95% CI 1.65, 7.01, *P* = 0.001).

## Antenatal CMV viremia is associated with infant CD8[+] T-cell immunophenotype

Given the effects of maternal CMV co-infection on infant mortality, we reasoned that CMV exposure may also influence infant immunophenotype. CMV is a highly immunostimulatory virus, with particular

**Table 3 | Associations between antenatal cytomegalovirus DNAemia and child mortality through 18 months**

| Antenatal cytomegalovirus DNAemia | Child mortality, n/N (%) | Hazard ratio | P | Adjusted hazard ratio[a] | P |
|---|---|---|---|---|---|
| Undetectable | 20/396 (5.1%) | Reference | | Reference | |
| Detectable | 20/175 (11.4%) | 2.29 (1.21, 4.31) | 0.010 | 2.11 (1.09, 4.09) | 0.027 |
| Undetectable | 20/396 (5.1%) | Reference | | Reference | |
| Detectable <45 copies/mL | 12/137 (8.8%) | 1.79 (0.90, 3.58) | 0.099 | 1.69 (0.82, 3.49) | 0.155 |
| ≥45 copies/mL | 8/38 (21.1%) | 4.13 (1.81, 9.43) | 0.001 | 3.32 (1.43, 7.72) | 0.005 |
| Log rise in CMV viral load | N/A | 1.73 (1.20, 2.51) | 0.004 | 1.62 (1.11, 1.36) | 0.013 |

Cox proportional hazards models were used to evaluate associations. Presented P-values have not been adjusted for multiple comparisons.
[a]Adjusted for maternal HIV viral load, CD4 count, gestational age at blood sampling and randomised trial arm.

effects on CD8[+] T-cell populations[30], and CD8[+] T-cell expansion among adults living with HIV is strongly associated with CMV co-infection[31]. We, therefore, hypothesised a role for CMV exposure in driving the higher infant CD8[+] T-cell activation, differentiation, proliferation, senescence and exhaustion we found in HEU vs. HIV-unexposed infants.

Among HIV-positive women with available infant immunophenotyping data, we identified CMV viremia in 32/102 (31%) women in the second trimester, and 20/85 (24%) in the third trimester. CMV viraemia in the third trimester was strongly associated with higher proportions of differentiated, senescent and exhausted infant CD8[+] T-cells after adjusting for plausible confounders including maternal HIV viral load (Table 4). HEU infants with third-trimester CMV exposure tended to have reductions in naïve CD8[+] T-cells ($P = 0.011$) and correspondingly higher percentages of memory cells, together with more CD28[−]/CD57[+] senescent CD8[+] T-cells ($P < 0.001$), and more CD8[+] T-cells expressing PD-1 ($P = 0.001$) (Table 4). There were no associations between CMV viraemia at 32-gestational weeks and proportions of infant-activated or proliferating CD8[+] T-cells (Table 4).

Given the associations between infant immune activation and both HIV exposure and third-trimester CMV exposure, we built a multivariable model to understand the interaction between the two exposures, adjusting for potential confounding factors (time of maternal and infant blood sampling, infant sex, and randomised trial arm). Maternal HIV was independently associated with proportions of infant-activated and proliferating CD8[+] T-cells. Both maternal HIV and CMV were independently associated with proportions of infant differentiated, senescent, and exhausted CD8[+] T-cells; the difference in CD8[+] T cells proportions was greater between CMV-exposed vs. CMV-unexposed than for HIV-exposed vs. HIV-unexposed infants (Table 5).

## HEU infants have earlier acquisition of CMV, which is associated with immune development

In contrast to high-income countries, where CMV is generally acquired in adolescence or adulthood, CMV is commonly acquired in infancy in sub-Saharan Africa[32]. Therefore, after finding high antenatal exposure to replicating CMV virus, particularly in the context of maternal HIV, we hypothesised that earlier infant acquisition of CMV infection in HEU compared to HIV-unexposed infants may drive differences in HEU infant immunophenotypes. Among infants with immunophenotyping data, 89 HEU and 100 HIV-unexposed infants had saliva samples tested for CMV by DNA PCR at a median age of 104 days (IQR 98, 118). Overall, 74% of HEU infants vs. 48% of HIV-unexposed infants had CMV detected in saliva at 3 months of age (adjusted odds ratio 3.00, 95% CI 1.56, 5.76; $P = 0.001$). On adjusted analyses, third trimester, but not the second trimester, perinatal CMV exposure was associated with early CMV acquisition amongst infants (Supplementary Table 5). HEU infants with CMV acquisition by 3 months, vs. those without CMV, had significantly higher percentages of CD38[+]/HLADR[+] activated CD8[+] T-cells ($P < 0.001$, Fig. 3A), Ki67[+] proliferating CD8[+] T-cells ($P = 001$, Fig. 3B); a more differentiated CD8+ cell phenotype (Fig. 3F−I); and more CD28[−]/CD57[+] senescent cells ($P < 0.001$) and PD-1[+] exhausted cells ($P < 0.001$) (Fig. 3C−E and Table 4). Among HIV-unexposed infants, early CMV

acquisition was also associated with CD8[+] T-cell immunophenotype, although the magnitude of association was less (Supplementary Table 6). Remarkably, HEU infants with expansive proportions of highly differentiated (effector memory and TEMRA), senescent and exhausted cells were confined solely to those with early CMV acquisition (Fig. 3).

Putting these findings together, two plausible explanations for the immune profiles of HEU infants should be considered. First, the primary driver of altered CD8[+] T-cell phenotype is earlier CMV acquisition in HEU infants compared to HIV-unexposed infants and greater CD8[+] T-cell differentiation and exhaustion in response to CMV in those infants with HIV exposure. Second, CD8[+] T-cell activation in HEU infants leads to susceptibility to earlier CMV acquisition. Our findings therefore add to the developing literature on the interaction between HIV and CMV[27], further highlighting CMV as a critical co-infection in mothers and infants affected by HIV.

## The relative associations of maternal inflammation and CMV-related immune activation differed by infant sex

Within HIV-exposed infants, we noticed stark differences in the relationship between maternal exposures and infant mortality by sex. Maternal CRP was strongly associated with mortality of HIV-exposed girls but not HIV-exposed boys, and exposure to CMV viremia was more strongly associated with mortality of HIV-exposed boys but not girls. To explore these differences further, we built multivariable models with both exposures, adjusting for potential confounding factors (maternal HIV viral load, CD4 count, timing of blood sampling, and randomised trial arm). In girls, mortality was more closely associated with CRP than CMV; in boys, mortality was more closely associated with CMV than CRP (Table 6).

Alterations in immunophenotype and biomarkers of systemic inflammation between HEU and HIV-unexposed infants also differed by infant sex, with differences generally of greater magnitude in boys compared to girls. When we disaggregated analyses by sex, we found that the lower CRP concentrations in HEU vs. HIV-unexposed infants were mostly driven by differences between boys (adjusted Tobit $\beta$ 0.45, 95% CI 0.19, 1.01) rather than girls (adjusted Tobit $\beta$ 0.65, 95% CI 0.27, 1.53). For boys, there was some evidence that IL-6 was lower in HEU compared to HIV-unexposed infants (Tobit $\beta$ 0.83, 95% CI 0.28, 2.49), while for girls, there was some evidence that IL-6 was higher in HEU compared to HIV-unexposed infants (Tobit $\beta$ 1.74, 95% CI 0.51, 5.92) (Supplementary Table 7). Differences in monocytes between groups were stronger in boys than girls, mostly driven by an expansion in non-classical, rather than intermediate, monocytes (Supplementary Table 8). Differences in CD8[+] T-cell development between HEU and HIV-unexposed infants were more pronounced in boys (Supplementary Table 8). Similarly, the alterations in immunophenotype associated with perinatal CMV exposure were more marked in boys than girls (Supplementary Table 9).

## Discussion

We aimed to define the biological pathways arising from antenatal HIV exposure that impact infant outcomes in early life, and our study has

**Table 4 | Differences in T-cell and monocyte phenotypes in HIV-exposed uninfected children at 1 month of age with and without third-trimester CMV exposure and early-life CMV acquisition**

| | Third-trimester CMV exposure | | | | Early CMV acquisition | | | |
| | Mean percentage (95% CI) | | | | Mean percentage (95% CI) | | | |
| | Detectable CMV viraemia (N = 20) | Undetectable CMV viraemia (N = 65) | Adjusted[a] coefficient (95% CI) | P | Early CMV acquisition (N = 53) | CMV negative 3 months (N = 18) | Adjusted[a] coefficient (95% CI) | P |
|---|---|---|---|---|---|---|---|---|
| Naïve CD8+ | 73.91 (60.85, 86.96) | 86.35 (81.22, 91.47) | −0.60 (−1.07, −0.14) | 0.011 | 79.41 (72.32, 86.51) | 96.79 (94.65, 98.92) | −0.97 (−1.34, −0.60) | <0.001 |
| Central memory CD8+ | 12.93 (6.69, 19.17) | 8.81 (5.27, 12.36) | 0.27 (−0.08, 0.61) | 0.135 | 11.48 (7.12, 15.84) | 2.97 (0.96, 4.99) | 0.62 (0.32, 0.92) | <0.001 |
| Effector memory CD8+ | 5.16 (1.37, 8.96) | 1.64 (0.43, 2.86) | 0.62 (0.16, 1.07) | 0.008 | 3.31 (1.51, 5.11) | 0.05 (0.00, 0.11) | 1.49 (0.80, 2.17) | <0.001 |
| TEMRA CD8+ | 8.01 (2.16, 13.86) | 3.19 (1.30, 5.08) | 0.67 (0.21, 1.12) | 0.004 | 5.80 (2.84, 8.77) | 0.18 (0.05, 0.31) | 1.27 (0.95, 1.59) | <0.001 |
| CD28+ CD57- CD8+ | 74.46 (63.33, 85.58) | 86.66 (82.31, 91.01) | −0.57 (−0.96, −0.18) | 0.004 | 77.50 (71.38, 83.61) | 96.77 (95.18, 98.36) | −1.01 (−1.33, −0.69) | <0.001 |
| CD28- CD57+ CD8+ | 14.13 (6.34, 21.92) | 4.06 (1.85, 6.27) | 0.85 (0.44, 1.27) | <0.001 | 9.43 (5.40, 13.46) | 0.33 (0.13, 0.54) | 1.31 (0.99, 1.62) | <0.001 |
| PD-1+ CD8+ T-cells | 11.15 (4.30, 18.00) | 4.65 (2.38, 6.91) | 0.68 (0.29, 1.08) | 0.001 | 8.09 (3.35, 11.73) | 1.27 (0.82, 1.73) | 0.74 (0.47, 1.02) | <0.001 |
| CD38+ HLA-DR+ CD8+ | 14.92 (8.24, 21.60) | 11.05 (6.90, 15.21) | 0.32 (−0.05, 0.68) | 0.089 | 16.68 (11.34, 22.03) | 2.27 (1.23, 3.31) | 0.92 (0.62, 1.22) | <0.001 |
| Ki67+ CD8+ | 16.79 (8.04, 25.55) | 14.95 (9.61, 20.29) | 0.10 (−0.32, 0.53) | 0.631 | 18.84 (12.54, 25.13) | 6.08 (3.14, 9.02) | 0.69 (0.29, 1.08) | 0.001 |
| Classical monocytes | 59.19 (52.45, 65.93) | 56.28 (52.56, 60.01) | 0.05 (−0.13, 0.22) | 0.612 | 57.56 (53.70, 61.43) | 59.86 (53.78, 65.93) | 0.02 (−0.20, 0.23) | 0.890 |
| Intermediate monocytes | 15.84 (10.34, 21.34) | 11.37 (9.58, 13.16) | 0.28 (0.05, 0.51) | 0.017 | 10.87 (9.01, 12.73) | 17.95 (12.42, 23.48) | −0.26 (−0.63, 0.10) | 0.159 |
| Non-classical monocytes | 24.97 (19.84, 30.11) | 32.34 (27.91, 36.78) | −0.23 (−0.42, −0.03) | 0.023 | 31.57 (27.06, 36.07) | 22.19 (16.28, 28.11) | 0.12 (−0.17, 0.41) | 0.429 |

Compared using fractional regression. All statistical tests are two-sided, and presented P-values have not been adjusted for multiple comparisons.
[a]Adjusted for maternal HIV viral load/CD4 count, gestational age at the time of maternal blood sampling, exact infant age in days at the time of infant blood sampling, infant sex, randomised trial arm.

five main findings. First, systemic inflammation among women with HIV, as measured by CRP, is strongly associated with infant mortality, suggesting that interventions targeting maternal inflammation and its causes during pregnancy might reduce infant mortality. Second, compared to infants who are not HIV-exposed, HEU infants have differences in immune development, with higher proportions of differentiated, activated, proliferating, senescent and exhausted CD8+ T-cells, higher sCD14 concentrations, and altered proportions of monocyte subsets. Third, these differences are particularly pronounced in boys compared to girls, highlighting sex-specific differences in immune development among infants with HIV exposure. Fourth, maternal CMV viraemia during pregnancy is more common in women with HIV, and is independently associated with infant mortality. Fifth, CMV exposure during pregnancy and acquisition during infancy are more common in HEU infants, and has a striking association with the CD8+ compartment, suggesting it may be the major driver of T-cell perturbations in HEU infants. Collectively, these findings show how the skewed immune milieu of women with HIV in pregnancy—characterised by inflammation, immune dysfunction and co-infections—shapes immune development in their offspring.

Despite >80% coverage of antiretroviral therapy, representative of present-day Zimbabwe[1], maternal biological characteristics were associated with infant mortality through 18 months of age, and associations remained apparent in adjusted models. We demonstrate for the first time that maternal CRP in early pregnancy is independently associated with mortality of HIV-exposed infants, but not HIV-unexposed infants. This finding has two public health implications. First, because CRP is inexpensive and simple to measure, antenatal point-of-care testing could be utilised at antenatal booking to identify those most at risk of infant mortality, to enable differentiated care for high-risk pregnancies. Although early HIV diagnosis, ART initiation and virological suppression remain important aspects of care, future studies should evaluate the utility of CRP as an additional measure of infant risk stratification. Second, antenatal interventions beyond ART that specifically target inflammation may be required to improve the survival of infants born to mothers with HIV. The drivers of maternal inflammation remain uncertain but could include microbial and vascular causes; there is a known association between CRP and bacterial and fungal infections, which are more common in HIV infection, and there remains a complex interplay between malnutrition, infection, inflammation, and impaired placental blood flow[33].

To further explore the potential biological mechanisms of adverse clinical outcomes among HIV-exposed infants, we next characterised cellular immunophenotype and soluble immune biomarkers in blood samples from HEU infants. Overall we identified evidence of immune activation in HEU compared to HIV-unexposed infants, which is similar to recent findings in SARS-CoV-2, where exposed but uninfected neonates had abnormalities of immunophenotyping, including higher percentages of natural killer cells and regulatory T-cells, higher plasma cytokines, and exaggerated cytokine production by neonatal immune cells[34]. Soluble CD14 is strongly associated with disease progression in adults with chronic HIV infection[35], was significantly higher in HEU compared to HIV-unexposed infants, and was accompanied by lower proportions of the least mature classical monocyte subset in favour of more mature intermediate and non-classical subsets. The most noticeable difference in immunophenotype was CD8+ T-cell activation in HEU infants, with a more activated and mature but senescent CD8+ cell phenotype. These findings may reflect greater exposure to pathogens in early life in HEU infants, but whether immune activation and differentiation is a helpful response to multiple microbial exposure or a detrimental exaggerated response, is uncertain. A strong correlation between proportions of memory T-cell cells and markers of T-cell senescence and exhaustion may represent a defect in maturation, whereby CD8+ cells incompletely differentiate, as in HIV infection[23,36]. Therefore, despite HEU infants having higher proportions of memory

**Table 5 | Associations between HIV exposure, CMV exposure in the third trimester and infant immune activation at 1 month of age**

| | Coefficient for HEU vs. HIV-unexposed children, adjusted for 32-week CMV status[a] | P | Coefficient for CMV-exposed vs. CMV-unexposed children, adjusted for HIV status[a] | P |
|---|---|---|---|---|
| Naïve CD8[+] T-cells | −0.30 (−0.57, −0.04) | 0.025 | −0.40 (−0.73, −0.07) | 0.017 |
| Central memory CD8[+] T-cells | 0.22 (−0.02, 0.47) | 0.067 | 0.21 (−0.08, 0.49) | 0.153 |
| Effector memory CD8[+] T-cells | 0.34 (0.05, 0.64) | 0.021 | 0.47 (0.12, 0.82) | 0.008 |
| TEMRA CD8[+] T-cells | 0.23 (−0.07, 0.53) | 0.132 | 0.42 (0.10, 0.74) | 0.011 |
| CD28[+] CD57[−] CD8[+] T-cells | −0.31 (−0.54, −0.07) | 0.011 | −0.35 (−0.64, −0.06) | 0.018 |
| CD28[−] CD57[+] CD8[+] T-cells | 0.30 (0.03, 0.57) | 0.032 | 0.46 (0.15, 0.77) | 0.003 |
| PD-1[+] CD8[+] T-cells | 0.19 (−0.07, 0.44) | 0.154 | 0.35 (0.06, 0.64) | 0.019 |
| CD38[+] HLA-DR[+] CD8[+] T-cells | 0.36 (0.10, 0.61) | 0.007 | 0.21 (−0.07, 0.50) | 0.146 |
| Ki67[+] CD8[+] T-cells | 0.27 (0.02, 0.53) | 0.036 | 0.21 (−0.11, 0.53) | 0.199 |
| Classical monocytes | −0.12 (−0.25, 0.00) | 0.048 | 0.04 (−0.12, 0.21) | 0.629 |
| Intermediate monocytes | 0.03 (−0.09, 0.15) | 0.654 | 0.12 (−0.05, 0.29) | 0.156 |
| Non-classical monocytes | 0.13 (−0.03, 0.28) | 0.111 | −0.12 (−0.33, 0.09) | 0.248 |

Compared using fractional regression. All statistical tests are two-sided, and presented P-values have not been adjusted for multiple comparisons.
[a]Also adjusted for time of maternal and infant blood sampling, infant sex, and randomised trial arm. *CMV* cytomegalovirus.

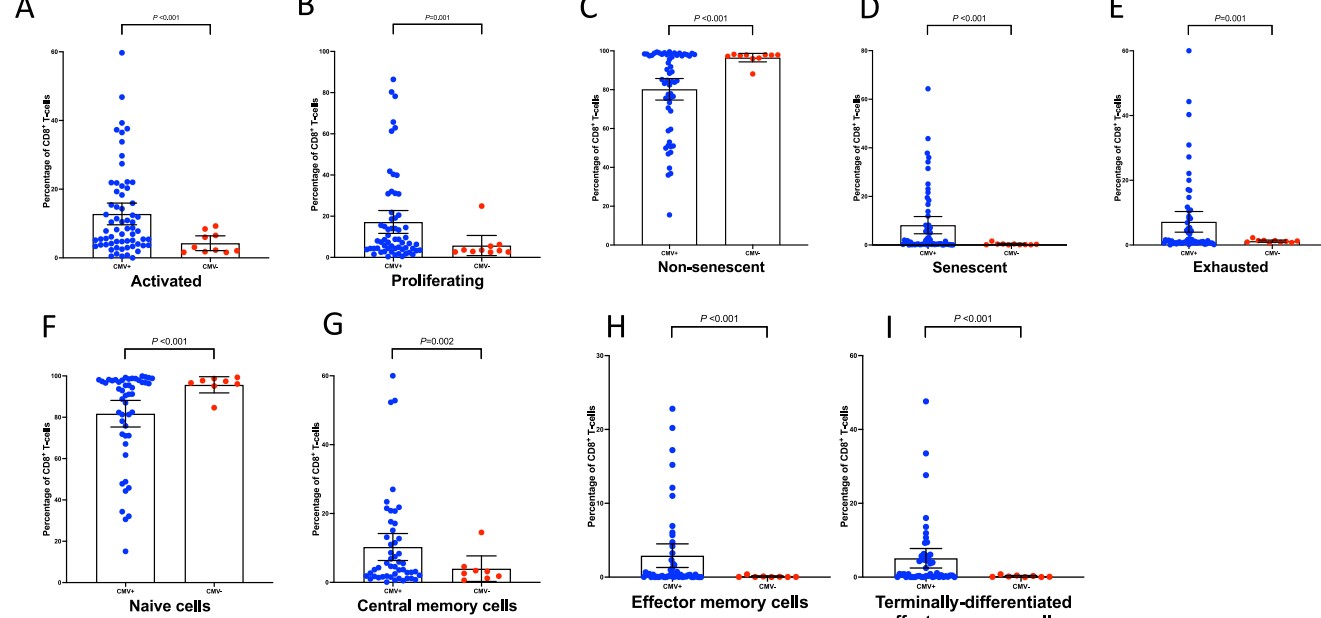

**Fig. 3 | CD8+T-cell activation in HEU children with and without early CMV acquisition.** Bars at mean with 95% confidence intervals. Compared using fractional regression. P-values adjusted for infant sex, exact age at the time of blood and saliva collection and randomised trial arm. All statistical tests are two-sided, and presented P-values have not been adjusted for multiple comparisons. **A** comparison of CD8[+] T-cell activation between HEU children with vs. without early CMV acquisition; **B** comparison of CD8[+] T-cell proliferation between HEU children with vs. without early CMV acquisition; **C, D** comparison of CD8[+] T-cell senescence between HEU children with vs. without early CMV acquisition; **E** comparison of CD8[+] T-cell exhaustion between HEU children with vs. without early CMV acquisition; **F–I** comparison of CD8[+] T-cell differentiation between HEU children with vs. without early CMV acquisition. Activated: CD38[+]/HLA-DR[+]; proliferating: Ki67[+]. Non-senescent: CD28[+]/CD57[−]; senescent: CD28[−]/CD57[+]; exhausted: PD-1[+]. Bars at mean with 95% confidence intervals. HEU: HIV-exposed uninfected. Blue: HEU children with early acquisition of CMV; red: HEU children without early acquisition of CMV. Source data are provided as a Source Data file.

cells than HIV-unexposed infants, an abundance of CD8[+]/CD27[−]/CD28[−]/CD57[+] or CD8[+]/CD27[−]/PD-1[+] central memory and effector memory cells may have poorer cytolytic function and less cytokine secretory potential[23,37,38] than is needed to successfully handle pathogens. Therefore, although the differentiation process of T-cells is a critical aspect of antiviral immunity, particularly given that memory T-cells provide faster responses to future microbial exposures[39,40], the differentiated phenotype in HEU infants should be viewed in the context of concurrent reductions in CD28 and gains in CD57 and PD-1 expression.

T-cell senescence may also have broader effects on immune homeostasis and defence against pathogens. For example, CD57[+] CD8[+] T-cells have been shown to suppress immunoglobulin production, and this T-cell-mediated immunosuppressive effect is enhanced in HIV infection[39,41]. Overall, the immunophenotype in HEU infants suggests a dysregulation in CD8[+] T-cell differentiation; rapid over-stimulation could be the result of exaggerated responses or exposure to excessive antigens leading to a mature but senescent and exhausted phenotype. It is therefore plausible that these differences in immunophenotype lead

**Table 6 | Adjusted hazard ratios for mortality adjusted for mortality of children born to mothers with HIV according to infant sex**

| | All children | | Male | | Female | |
|---|---|---|---|---|---|---|
| | Adjusted[a] hazard ratio (95% CI) | P | Adjusted[a] hazard ratio (95% CI) | P | Adjusted[a] hazard ratio (95% CI) | P |
| Maternal CRP below median | Reference | | Reference | | Reference | |
| Maternal CRP at or above the median | 3.29 (1.66, 6.52) | 0.001 | 2.31 (0.85, 6.26) | 0.101 | 5.23 (1.60, 17.09) | 0.006 |
| CMV undetectable | Reference | | Reference | | Reference | |
| CMV < 45 copies/mL | 1.56 (0.77, 3.17) | 0.220 | 1.55 (0.50, 4.81) | 0.451 | 1.54 (0.56, 4.20) | 0.403 |
| CMV ≥ 45 copies/mL | 3.49 (1.45, 8.42) | 0.005 | 5.86 (1.72, 19.99) | 0.005 | 1.65 (0.52, 5.30) | 0.398 |

Cox proportional hazard models were used to evaluate associations. Presented P-values have not been adjusted for multiple comparisons.
[a]Adjusted for maternal HIV viral load, CD4 count, timing of blood sampling, and randomised trial arm.

to impaired handling of common infections and subsequent morbidity and mortality in infancy.

In the SHINE cohort, around one-third of pregnant women with HIV had detectable CMV viraemia in pregnancy, and this was independently associated with a two-to-three-fold increase in infant mortality through 18 months of age. CMV viraemia was mostly low-level (<45 copies/mL), but higher antenatal CMV viral loads were associated with greater infant mortality. Although CMV viremia could reflect more severe HIV disease severity in pregnant women, this is unlikely to be the full explanation because the association was independent of maternal HIV viral load and CD4 count. There are several other plausible explanations, and these are likely to be multifactorial, including an impact of CMV exposure on birth outcomes such as low birthweight, small-for-gestational age, and prematurity. We demonstrated strong correlations between late gestational exposure to maternal CMV and infant CD8[+] T-cell differentiation, senescence and exhaustion only in HEU infants, suggesting a specific effect of maternal CMV in the context of HIV. Early infant acquisition of CMV was also strongly associated with CD8[+] T-cell phenotype, particularly in HIV-exposed infants. Three findings therefore highlight CMV as a critical co-infection in mothers and infants affected by HIV: first, maternal CMV viremia is more common in women with HIV; second, HEU infants acquire CMV earlier in infancy than HIV-unexposed infants; and third, perinatal CMV exposure is more closely associated with immune phenotype in HEU infants than it is in HIV-unexposed infants. CMV has previously been identified as an environmental factor that can shape long-term differences in immune cell phenotype and systemic immune biomarkers between healthy twins discordant for CMV in high-income settings[42,43]; thus, early discrepancies in CMV acquisition between HEU and HIV-unexposed infants could impact immune development across the life-course. Congenital CMV has been reported to be higher in HIV-exposed infants[44], although we could not distinguish congenital vs. postnatal CMV in this cohort. Interventions to reduce CMV reactivation or reinfection in pregnant women with HIV may have benefits for their HIV-exposed infants.

We found sex-specific differences in immune development. In general, female fetuses tend to have enhanced adaptability to intrauterine stress, while male fetuses tend to have increased systemic inflammation and higher cytokine concentrations[45,46]. Blood immune cells from male compared to female children produce higher concentrations of inflammatory cytokines in response to in vitro antigen stimulation[45,47], in keeping with our findings of exaggerated responses to viral infections. It is therefore consistent with the existing literature on immune ontogeny that we found exaggerated differences in immune activation associated with HIV exposure in boys. In the larger SHINE cohort, mortality and stunting were more common in boys[8,48,49], highlighting the particular vulnerability of male infants in this setting. Finally, we found that mortality in HIV-exposed boys is more closely related to their virus-related altered immunophenotype (from HIV and CMV exposure), whereas mortality in HIV-exposed girls is more closely related to maternal inflammatory pathways. Exploring the differential responses

between boys and girls may help to identify underlying disease mechanisms, which would inform future intervention strategies.

This study has strengths and limitations. A large cohort with rich data allowed detailed evaluations of the associations between maternal biomarkers and infant immunophenotype and mortality in a typical rural African context. Infants were universally breastfed, and samples for the immunological/intestinal analyses were collected before weaning took place, meaning they were not biased by differences in feeding or weaning between groups. However, these analyses were not the primary aims of the original trial. Laboratory analyses were reliant on sample collection, availability and storage. Bias could have been introduced from the mothers and infants with available samples, and technical factors relating to transportation and long-term cryopreservation may have affected laboratory assays. By design, infants who died prior to blood sampling at 1 month had no immunophenotyping data. Similarly, we could not ascertain HIV acquisition in neonates who died, meaning it is unclear whether some clinical associations are due to vertical HIV transmission or poor clinical outcomes among HEU infants. However, all immunological evaluations were undertaken in infants confirmed to be HIV-exposed but uninfected.

In summary, we show biological associations between perinatal HIV/CMV exposure, inflammation and mortality through 18 months of age. Maternal systemic inflammation and cytomegalovirus co-infection may be novel targets for interventions during pregnancy. In the goal to eliminate paediatric HIV infection, we must also ensure that the growing population of infants who are HIV-exposed but uninfected not only survive but thrive and enjoy healthy and fulfilling lives[5].

## Methods

### Ethical approvals

The Medical Research Council of Zimbabwe (MRCZ/A/1675), Johns Hopkins Bloomberg School of Public Health (JHU IRB # 4205) approved the study protocol. All mothers provided written informed consent to join the study. The SHINE trial is registered at Clinical-Trials.gov (NCT01824940).

### Study population

Participants in this study were recruited to the Sanitation Hygiene Infant Nutrition Efficacy (SHINE) trial, a 2×2 factorial cluster randomised trial assessing the effects of improved infant and young infant feeding (IYCF) and improved water, sanitation, and hygiene (WASH) on stunting and anaemia at 18 months of age[50]. SHINE recruited pregnant women from two rural Zimbabwean districts with 15% antenatal HIV prevalence between November 2012 and March 2015[50]. Breastfeeding was universal, and there was a high prevalence of exclusive breastfeeding (80–90% between 1-3 months of age, Mbuya et al.[51]. Mother-infant pairs lived in clusters randomised to standard of care (SOC); IYCF (20 g small-quantity lipid-based nutrient supplement/day from 6 to 18 months of age, and complementary feeding counselling); WASH (pit latrine and 2 hand-washing stations, liquid soap and chlorine, a clean play space, and hygiene counselling); or IYCF + WASH (all

interventions)[50]. Trial analyses were stratified by maternal HIV status, and the primary endpoints have been reported previously[48,49,52,53]. For mortality analyses, all liveborn infants were included (Supplementary Table 1). For immunological analyses, all infants in the SHINE trial born to women living with HIV were included if they were confirmed HIV-negative at 18 months of age, had length measured at 18 months of age, had laboratory samples available from the 1-month postpartum visit, and had mothers with available laboratory samples from the trial baseline visit. From the pool of infants born to women without HIV, infants were eligible for inclusion if they had length measured at 18 months of age, had laboratory samples available from the 1-month visit, and had mothers with available laboratory samples from the trial baseline visit. One infant was chosen for each HIV-exposed uninfected infant, matched on trial arm, sex, and season of birth. If more than one HIV-unexposed infant met the matching criteria, one of the infants was chosen at random. Infants who died or were HIV-infected were excluded (Supplementary Table 10 and Supplementary Fig. 1).

## Clinical data collection
Research nurses visited mothers in early pregnancy (-2 weeks after consent) and at 32-gestational weeks, to assess maternal and household characteristics. At baseline, mothers had height, weight, and mid-upper arm circumference (MUAC) measured, and household wealth assessed[50]. Infant birth date, weight, sex and delivery details were transcribed from health records. Home visits were scheduled at 1, 3, 6, 12, and 18 months postpartum, and biological samples were collected. At postnatal visits, mothers received a small non-monetary gift to thank them and their infants for their time in participating in data collection. Mortality was assessed through home visits, village health worker reports, and telephone calls, with the date of death recorded where available. Maternal ART use was documented at 3 timepoints (baseline, 32-gestational weeks, and 1 month postpartum) based on maternal reports and a review of handheld medical records.

## Maternal HIV testing
At baseline and 32-gestational week visits, (as soon as possible thereafter within the allowable visit window), blood was collected from pregnant women into endotoxin-free EDTA tubes (BD Biosciences). Mothers were tested during pregnancy using a rapid test algorithm (Alere Determine HIV-1/2 test, and, if positive, INSTI HIV-1/2 test [Bio-Lytical Laboratories]). Women testing positive had CD4 cell counts measured (Alere Pima Analyser) and were referred to local clinics. Viral loads were not measured in real-time. National PMTCT guidelines changed from World Health Organization (WHO) Option B (maternal ART from 14 gestational weeks until the end of breastfeeding) to Option B+ (lifelong ART for all pregnant and breastfeeding women) in November 2013. Women enrolled in SHINE were encouraged to initiate co-trimoxazole and ART, to exclusively breastfeed, and to attend a local clinic at 6 weeks postpartum for early infant diagnosis and co-trimoxazole prescription. The first-line ART regimen in Zimbabwe was 2 nucleoside reverse transcriptase inhibitors (NRTIs) and a non-nucleoside reverse transcriptase inhibitor (NNRTI); the second-line regimen was 2 NRTIs and a protease inhibitor.

## Biological samples
Women living with HIV and enrolled in the SHINE study were invited to enroll in a sub-study in which infants had blood collected at 1, 3, 6, 12, and 18 months for HIV testing; infants of mothers not enrolled in the sub-study were only tested at 18 months. Infants testing positive were referred to local clinics for ART. Infants born to women living with HIV and testing HIV-negative at 18 months were classified as HEU. Infants not tested at 18 months due to caregiver refusal, defaulted visits, or loss to follow-up were classified as HIV unknown. Infants who died were classified based on their HIV test result at the last trial visit before death, or as HIV unknown if they had never been tested or were not tested at the last trial visit before death. Before 18 months of age, HIV was diagnosed using DNA polymerase chain reaction (PCR) on dried blood spot samples or RNA PCR on plasma; and after 18 months, by PCR or rapid test algorithm, depending on samples provided. Inconclusive or discordant results were retested to confirm status; if no further samples were available or repeat testing was inconclusive, infants were classified as HIV unknown. For infants at 1 month of age (as soon as possible thereafter within the allowable visit window of up to 90 days), up to 2.7 mL blood was collected from infants into endotoxin-free EDTA tubes (BD Biosciences). Saliva was collected at 3, 6, 9, 12 and 18 months of age by placing an oral soft swab under the tongue for 1-2 min. The whole swab was placed into a storage tube and frozen at −80C. From mid-2014, mother-infant pairs were invited to join a sub-study to investigate biomarkers of EED. Women were informed about the EED sub-study at their 32-week gestation visit and those with live births were enrolled at the 1-month postnatal visit, or as soon as possible thereafter. From children in the EED sub-study, biological specimens were collected at each postnatal visit (1, 3, 6, 12 and 18 months of age) to measure a range of EED biomarkers (stool neopterin, myeloperoxidase, alpha-1 anti-trypsin and REG-1β and plasma I-FABP[22,54], Stool was collected into plain tubes on the morning of the visit and stored at −80C. Blood was collected into EDTA tubes and centrifuged to collect plasma and buffy coat cells, which were stored at −80 °C.

Viral nucleic acid was extracted from 200 µL of plasma using the Abbott RealTime HIV-1 extraction assay. Each extraction included positive and negative controls. HIV was detected by real-time PCR on the Abbott m2000rt platform using the Abbott RealTime HIV Amplification Reagent Kit. The limit of detection of the assay was 150 copies/mL. Each amplification run included positive and negative controls. For CMV, viral nucleic acid was extracted from 100 µL of plasma or 50 µL of saliva using the QIAamp DSP Virus Spin Kit. Each extraction included positive and negative controls. Cytomegalovirus was detected by real-time PCR on the Abbott m2000rt platform using the Abbott RealTime CMV Amplification Reagent Kit, modified for the nucleic acid extraction method used, which therefore provided a limit of quantification of 45 copies/mL for plasma and 90 copies/mL for saliva. To limit false positives from maternal breast milk residue, infant saliva was collected at least 20 min after the last breastfeed, and positive saliva samples <450 copies/mL were considered negative ($N = 15$). Each amplification run included positive and negative controls.

## Systemic inflammatory markers
CRP, IL-6 and soluble CD14 were measured in maternal and infant plasma by ELISA using R&D Systems Quantikine kits according to the manufacturer's instructions. Soluble CD163 was measured in maternal and plasma by ELISA using R&D Systems DuoSet kits according to the manufacturer's instructions. I-FABP was measured by ELISA in plasma (Hycult Biotechnology, Uden, The Netherlands). Stool samples were tested by ELISA for neopterin (GenWay Biotech Inc, San Diego, CA, USA), myeloperoxidase (Immundianostik, Bensheim, Germany), alpha-1 anti-trypsin (BioVendor, Brno, Czech Republic), and REG-1β (TECH-LAB Inc, Blacksburg, VA, USA).

## Immunophenotyping
Six-colour flow cytometry was used to phenotype CD4+ and CD8+ T-cells and monocytes using a BD FACSVerse flow cytometer. Buffy coats from centrifuged (2800 g for 5 min) EDTA samples collected from participants were mixed with 1.8 mL of BD FACS lysing solution and incubated in the dark for 10 min. After a second centrifugation for 10 min at 400 g, the supernatant was discarded followed by resuspension, and 1 mL cold freezing medium was added. The cryovial was stored at −80C in a *Mr. Frosty* device, and the following day the cryovials were transferred to cryoboxes for ongoing storage. Cryopreserved buffy coat cells from each participant were thawed, split into 4 aliquots, and labelled with 4 panels of fluorescently labelled antibodies to characterise T-cell

activation, differentiation, senescence, exhaustion and monocyte phenotype. Using a hierarchical gating strategy (Supplementary Figs. 2–5) applied relative to fluorescence minus-one (FMO) controls, we first identified single leucocytes on the basis of side- and forward-scatter, and identified T-cells using CD3-FITC (BD Biosciences) and then classified CD4$^+$ (CD4/APC-H7, BD Biosciences) and CD8$^+$ (CD8/PE-Cy7, BD Biosciences), T-cell differentiation (CD45RA/APC, BD Biosciences; CD27/PE, BD Biosciences; CD31/PerCP-Cy5.5, Biolegend), activation (CD38/PE, BD Biosciences; HLA-DR/APC, BD Biosciences), proliferation (intracellular Ki67-PerCP-Cy5.5, staining performed after membrane permeabilization), senescence (CD28/APC, BD Biosciences; CD57/PE, BD Biosciences) and exhaustion (PD-1/PerCP-Cy5.5, BD Biosciences). Using the same batches of cryopreserved cells, for monocytes, we first identified single leucocytes on the basis of side- and forward-scatter, then identified monocytes using an anti-human lineage cocktail (CD3/CD19/CD20/CD56/APC, Biolegend) and HLA-DR/PE-Cy7 (BD Biosciences), and then classified classical, intermediate and non-classical monocytes using CD14/PE (Biolegend) and CD16/APC-Cy7 (Biolegend). Median fluorescence intensities (MFI) of inflammatory (CD40/PerCP-Cy5.5, Biolegend), maturation (HLA-DR/PE-Cy7, BD Biosciences) and T-cell co-stimulatory (CD86/FITC, Biolegend) markers were quantified. Supplementary Table 11 shows the clones and manufacturers for each fluorochrome. All antibodies are commercially available and have undergone quality checks/validation by the companies. Each immunophenotyping panel was optimised prior to implementation by comparing antibody labelling of fresh and cryopreserved buffy coat cell samples, and antibodies were titrated to increase the sensitivity for the affected marker. FACSuite was used for data collection. FlowJo version 10.7.1 was used for flow cytometry gating. The gating of all samples was blinded to maternal and infant characteristics. Fluorescence-minus-one (FMO) controls were used to assure consistent gating with adjustment to individual samples. A representative example of the flow cytometry gating strategy is shown in Supplementary Figs. 2–5. Samples were excluded if <3000 CD3$^+$ T-cells were acquired or <1000 HLA-DR$^+$ monocytes were acquired.

## Statistical analyses

Baseline characteristics were compared between HIV-exposed and HIV-unexposed infants, or between women with and without HIV infection, using multinomial and ordinal regression models with robust variance estimation, and Somers' D for medians, to account for within-cluster correlation. Cox proportional hazard models were used to evaluate associations between maternal biomarkers at the baseline SHINE visit and infant mortality through 18 months. When the exact age of infant death was unknown (2 of 51 deaths; 4%), an estimated age of death was assigned using the conditional median among children who died beyond the timepoint when the child was last reported alive[8].

Generalised estimating equations (GEE) with robust variance estimation and an exchangeable correlation to account for clusters were used to evaluate the relationships between maternal biomarkers and infant acquisition of HIV and CMV. Linear regression was used to ascertain associations between maternal HIV viral load and CRP. HIV viral load, CD4 count, CRP and CMV viral load were log$_{10}$-transformed to improve kurtosis. When measurements were below the lower limit of quantification for the assay (HIV viraemia, CMV viraemia, CRP), a value was assigned using the following formula: assigned value=(assay lower limit of detection)/√2[55]. Using this approach, all women with reactive HIV tests but with HIV viraemia below the limit of quantification were assigned values of 106 copies/mL. Women with detectable CMV viraemia below the limit of assay quantification were assigned values of 32 copies/mL. Women without detectable CMV were assigned values of 0 copies/mL. CRP results below the limit of quantification were assigned values of 0.007 mg/L. Covariables for multivariable Cox regression models were developed using direct acyclic graphs (DAG) using *Dagitty* software version 3.0 (www.dagitty.net).

Minimal sufficient adjustment sets for estimating the total effects were used, with the addition of study-related factors (gestational age at time of blood sampling and randomised trial arm).

GEE with robust variance estimation and an exchangeable correlation to account for cluster was used to compare concentrations of biomarkers between groups, adjusted for randomised trial arm, infant sex and exact infant age at the 1-month study visit for infants or gestational age for maternal samples (as these may have been associated with outcome values), and accounting for within-cluster correlation. Soluble biomarkers (soluble CD14, soluble CD163, CRP, IL-6) and median fluorescent intensities of monocyte-surface markers (CD40, CD86, HLA-DR) were compared by linear regression. All linear regression models were fitted by GEE with an exchangeable correlation to account for study cluster. CRP and IL-6 were log$_{10}$-transformed to improve kurtosis, and Tobit regression models with robust variance estimation were used when values below the limit of detection of the assay exceeded 5%. Tobit regression models were also fitted by GEE with an exchangeable correlation to account for the cluster. Linear regression was used to ascertain correlations between biomarkers, adjusted for randomised trial arm, infant sex, and exact infant age at the 1-month study visit for infants or gestational age or for maternal samples (as these may have been associated with outcome values), and accounting for within-cluster correlation. Proportions of T-cell and monocyte subsets, defined by surface and intracellular markers, were compared using fractional regression. Multivariable models were created for all analyses to account for infant sex, the exact age of the infant or the gestational age of the mother at the time of blood collection, and randomised trial arm (as these may have been associated with outcome values). To account for multiple comparisons of phenotypic data, we used the Benjamini-Yekutieli procedure with a false discovery rate of 0.25 to adjust the level at which we reported findings as significant, which estimated a threshold of $P = 0.009$ or below; we therefore highlight phenotypic differences only when the $P$-value is $\leq 0.009$. Spearman correlations were used to ascertain associations between maternal and infant immune cell phenotypes. To determine the influence of infant sex, all analyses were repeated after disaggregation by infant sex. Logistic regression was used to estimate odds ratios for CMV acquisition in infants with maternal HIV and CMV exposure compared to those without exposure. Logistic regression models were also fitted by GEE with an exchangeable correlation to account for study cluster. Reported $P$-values are two-sided. Stata v. 17.0 was used for all analyses. Graphs were made using Stata v. 17.0 and GraphPad Prism v. 9.2.0.

## Reporting summary

Further information on research design is available in the Nature Portfolio Reporting Summary linked to this article.

## Data availability

The SHINE Trial data generated in this study have been deposited in the ClinEpiDB database under accession code https://clinepidb.org/ce/app/workspace/analyses/DS_0086998c2f/new/variables/EUPATH_0035127/EUPATH_0044124. The SHINE Trial data are available under controlled access for tracking data usage, access can be obtained by submitting an access request then data access will be granted immediately upon request submission. The raw SHINE Trial data are protected and are not available due to data privacy laws. The processed SHINE Trial data are available at https://clinepidb.org. Source data are provided in this paper.

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

## Acknowledgements

We thank all the mothers, babies, and their families who participated in the SHINE trial, the leadership and staff of the Ministry of Health and Child Care in Chirumanzu and Shurugwi districts and Midlands Province (especially environmental health, nursing, and nutrition) for their roles in operationalization of the study procedures, the Ministry of Local Government officials in each district who supported and facilitated field operations, Phillipa Rambanepasi and her team for proficient management of all the finances, Virginia Sauramba for management of compliance issues, and the programme officers at the Gates Foundation and the Department for International Development, who enthusiastically worked with us over a long period to make SHINE happen. Funding was received from the following organisations. Bill & Melinda Gates Foundation (OPP1021542 and OPP1143707). United Kingdom Department for International Development (DFID/UKAID). Wellcome Trust (203905/Z/16/Z (C.E.), 210807/Z/18/Z (C.E.), 093768/Z/10/Z (A.J.P.), 108065/Z/15/Z (A.J.P.), 206225/Z/17/Z (C.D.B.; Wellcome Trust/Royal Society-funded Sir Henry Dale Postdoctoral Fellowship). CIPHER programme of the International AIDS Society (grant number 2019/857-EVA) (C.E.). Swiss Agency for Development and Cooperation. US National Institutes of Health (2R01HD060338-06). UNICEF (PCA-2017-0002). The funders had no role in the design of the study and collection, analysis, and interpretation of data and in writing the manuscript.

## Author contributions

Conceptualised and study design: C.E., J.H.H., R.N., A.J.P. Data and biospecimen collection and management: K.M., F.D.M., N.V.T., R.N. Biospecimen processing and laboratory work: C.E., K.M., S.R., M.G., P.M., J.B., C.N., C.D.B. Data analysis and interpretation: C.E., B.C., E.K.G., P.K., C.D.B., J.H.H., R.N., A.J.P. Writing—original draft: C.E. Writing—reviewing and editing: all authors.

## Competing interests

The authors declare no competing interests.

## Additional information

Ceri Evans ®[1,2] ✉, Kuda Mutasa[1], Sandra Rukobo[1], Margaret Govha[1], Patience Mushayanembwa[1], Bernard Chasekwa[1], Florence D. Majo[1], Naume V. Tavengwa[1], Jonathan Broad[2], Christie Noble ®[2], Ethan K. Gough ®[3], Paul Kelly ®[2,4], Claire D. Bourke ®[1,2], Jean H. Humphrey[1,3], Robert Ntozini ®[1] & Andrew J. Prendergast ®[1,2,3]

[1]Zvitambo Institute for Maternal and Child Health Research, Harare, Zimbabwe. [2]Blizard Institute, Queen Mary University of London, London, UK. [3]Department of International Health, Johns Hopkins Bloomberg School of Public Health, Baltimore, MD, USA. [4]Tropical Gastroenterology & Nutrition Group, University of Zambia School of Medicine, Lusaka, Zambia. ✉e-mail: ceri.evans@qmul.ac.uk

