## [Peer Review File · Nature Communications]

REVIEWER COMMENTS

Reviewer #1 (Remarks to the Author):

This important paper from Evans and colleagues presents data from a large cohort of HIV exposed and unexposed infants to understand better how maternal inflammation and CMV coinfection shape infant outcomes and immune development. The study is notable and innovative in that it holistically brings together key concepts of co-infections, immune development, inflammation and activation and clinical outcomes in a single study; whereas these phenomena have previously been studied singly. This is also the largest study to date of HIV/CMV coinfections in African infants, and the sample size for the immunology studies is impressive and increases confidence in the validity of the study findings. Even though this is a nested study in an RCT, the methodology is sound, and the sampling plan, statistical analyses, and limitations are comprehensively presented and appear to be rigorous and repeatable. The conclusions are justified by the data presented. The key findings in this study are that maternal inflammation in late gestation, and CMV (maternal gestational viremia and early infant CMV acquisition) are strongly associated with early life infant mortality and with shaping the phenotype of infant monocytes and CD8 T cells in HEU children. Together these data strengthen the evidence base for developing CMV-specific interventions to improve outcomes for HEU infants in LMIC. Additionally, the authors find interesting effect modification by infant sex assigned at birth, which present additional avenues for mechanistic studies tying HIV exposure and CMV to infant outcomes.

The key limitations of the study are discussed by the authors, and include the nested nature of the study design, technical challenges of working with cryopreserved PBMC, and potential survival bias and sampling bias introduced by infant mortalities and potential differential loss to follow-up. My comments primarily relate to clarifications, and do not detract from my enthusiasm for this paper.

1. Figure 1: Figures C & E have been swapped
2. Flow cytometry data: sampling for the flow cytometry studies is described to be restricted to those with month 18 visits but the Tables present month 1 data- did the investigators examine cellular phenotypes at 18 months to see if differences between children with and without early CMV acquisition persist?
3. EED/intestinal injury biomarker studies: What timepoints were examined? There could be differences between early life (breastfeeding) and post-weaning when food insecurity impedes nutrition, growth, exacerbates inflammation.
4. Supplemental Figure 16: unclear how the last two sets of columns (both labelled "Female: Adjusted Hazard ratio") differ
5. There is no mention of accounting for multiple comparisons- with the large number of comparisons for the phenotype data it would be good to do some type of adjustment, maybe Holm-Bonferroni?

6. Page 23 line 11 type: first “death” in this sentence should be “age”

7. Since maternal HIV indicators were not associated with increased CRP, it would be interesting to investigate available maternal nutritional indicators (for example: weight gain in pregnancy, % body fat, food insecurity etc) to see if maternal malnutrition could be related to increased inflammation in pregnancy.

8. Was early infant CMV acquisition associated with mortality?

Reviewer #2 (Remarks to the Author):

The manuscript by Evans et al. reports novel findings on the role of the inflammation and CMV viremia during pregnancy in the mortality and immune development in HIV-exposed infants. The manuscript provides interesting insights into the mechanism by which CMV might affect HIV survival trajectory and highlights immunological differences in infant following in utero HIV exposure. This is a single country (Zimbabwe) study, and it nestles within a convenience cohort from other projects. The manuscript is well written and provides wealth of valuable data, however, some of the discussion points and conclusions, especially, regarding global policies on the management of pregnant women are premature and overstate the significance of these findings. The analyses in this study were not the primary aims of the original trial and could represent significant bias from the mothers and infants with available samples and data. Prospective clinical trials are needed to validate the results from this study and to explore the public health implications. Here are a few specific suggestions for the authors:

Abstract: Suggest specifying the inflammation in the following statement “We demonstrate independent associations between maternal HIV, inflammation and cytomegalovirus co-infection and infant immune development and mortality” – please, specify whether you associate observed inflammation with HIV. Given the lack of association of CRP elevation with HIV viremia and ART, lack of association of CRP with infant mortality in non-exposed infants and known association of CRP elevation with bacterial and fungal infections, it is worth to clarify the speculated nature of the inflammation among women with HIV.

Methods: Please, specify whether the information was available/collected on maternal co-morbidities beyond CMV and nutritional status. Please, clarify whether infants in the study were breastfed.

Results:

P 6 – lines 26-27 - Please, consider changing “We therefore questioned whether maternal inflammation during pregnancy is associated with infant mortality” to the “We hypothesized that”

P 10 – line 12 – how were the infants selected for the sampling of “117 HEU infants and 102 HIV-unexposed infants”?

Please, consider adding a figure with the study flow chart with the cohorts of infants and women.

In the study only 18% of women with HIV had significant CMV viremia. The numbers of CMV studies are low in relationship to infant markers – “Among HIV-positive women with available infant immunophenotyping data, we identified CMV viremia in 32/102 (31%) women in the second trimester, and 20/85 (24%) in the third trimester”.

Discussion:

On page 16, the authors call for the public health approach measuring CRP during pregnancy in women living with HIV. How will the detected viremia be addressed in resource-limited settings? And do authors propose to stratify infant care based on single or repeat maternal measurement of CRP during pregnancy? In the Discussion authors are already labeling the pregnancies with elevated CRP as “high-risk” based on this single study. The authors also propose antenatal interventions beyond ART, but do not specify which ones.

In the Discussion the HIV cure topic is not addressed, yet studies are looking at using CNV co-infection with HIV as a solution to immune therapy of HIV. Please, consider addressing.

The authors also speculate in regards of identified gender differences as following “exploring these differences may be essential in further designing interventions for this population”. I am unaware of any neonatal interventions that are gender/sex based and find this approach unethical.

Reviewer #3 (Remarks to the Author):

This is an important and significant paper in the field of CMV and HEU children and immune development. The authors explore the impact of maternal immune activation and CMV infection on the shape of infant immunity between HEU and HUU controls in Zimbabwe, using participants from the SHINE trial. The noteworthy results are that maternal CRP from HIV infected mothers is associated with infant mortality; HEU infant have activated and more mature CD8+ T cells along with activated monocyte populations; significant differences between boys and girls; CMV viral detection during gestation is independently associated with mortality; CMV exposure during pregnancy and acquisition in the first 3 months of life associate with CD8+ T cell activation and altered phenotypes. The MS was however, very difficult and annoying to read as the majority of tables did not correspond to the narrative results. This warrants a major revision.

Major Comments

1. Adverse birth outcomes, such as pre-term birth, are not factored into the analysis as a major driver in infant mortality. Is CMV detection in 2nd or 3rd trimester associated with PTB and poor infant survival? If this is unknown, there needs to be a mention of this in the discussion.

2. Is the type of ARV regimen impactful on maternal and HEU immune activation? Those on a TFV-based regimen are 397 and those on an AZT-based regimen are 120, so an ability to identify maternal ARV regimen as a driver or a confounder would be good to build in.

3. There is almost an endless list of tables (23 in total) and the authors need to prioritize which of these are central to their story and which are supplementary and which can be omitted entirely (or amalgamated). Table 24 seems to be a repeat of table 1 and table 25 is mislabeled. As mentioned above, very few tables correspond with the description in the results.

4. The title of the MS is based on inflammation and CMV, but CMV doesn't come into the picture until halfway through the results, where there is a long list of how maternal HIV impacts, or does not impact, on immune phenotypes in the HEU. The authors may wish to trim this down or consolidate the data.

Minor comments

1. Some of the compensation settings in Figure 4 look suspect – such as CD4 vs CD8, where CD8 is spilling into the CD4 channel. CD57 looks over-compensated. What was the QC/QA on the flow assays and were compensation beads used?

2. There is also no viability stain (ie. being able to discriminate live/dead). This is particularly concerning as spurious staining may result from antibodies non-specifically binding to dead cells. How did the authors account for this

3. What are the changes in the numbers of cells, only frequencies are given

4. Not sure sex differences is a supplementary table

5. Lines 25-27 on page 15 – this is a very tenuous connection (as correlations do not suggest cause) and should appear in the discussion and not the results

6. The authors could use an unbiased approach to the phenotype analysis, rather than manual gating.

REVIEWER COMMENTS

Reviewer #1 (Remarks to the Author):

This important paper from Evans and colleagues presents data from a large cohort of HIV exposed and unexposed infants to understand better how maternal inflammation and CMV coinfection shape infant outcomes and immune development. The study is notable and innovative in that it holistically brings together key concepts of co-infections, immune development, inflammation and activation and clinical outcomes in a single study; whereas these phenomena have previously been studied singly. This is also the largest study to date of HIV/CMV coinfections in African infants, and the sample size for the immunology studies is impressive and increases confidence in the validity of the study findings. Even though this is a nested study in an RCT, the methodology is sound, and the sampling plan, statistical analyses, and limitations are comprehensively presented and appear to be rigorous and repeatable. The conclusions are justified by the data presented. The key findings in this study are that maternal inflammation in late gestation, and CMV (maternal gestational viremia and early infant CMV acquisition) are strongly associated with early life infant mortality and with shaping the phenotype of infant monocytes and CD8 T cells in HEU children. Together these data strengthen the evidence base for developing CMV-specific interventions to improve outcomes for HEU infants in LMIC. Additionally, the authors find interesting effect modification by infant sex assigned at birth, which present additional avenues for mechanistic studies tying HIV exposure and CMV to infant outcomes.

The key limitations of the study are discussed by the authors, and include the nested nature of the study design, technical challenges of working with cryopreserved PBMC, and potential survival bias and sampling bias introduced by infant mortalities and potential differential loss to follow-up. My comments primarily relate to clarifications, and do not detract from my enthusiasm for this paper.

1. Figure 1: Figures C & E have been swapped

Response: Thank you very much for highlighting this, which has now been corrected.

2. Flow cytometry data: sampling for the flow cytometry studies is described to be restricted to those with month 18 visits but the Tables present month 1 data- did the investigators examine cellular phenotypes at 18 months to see if differences between children with and without early CMV acquisition persist?

Response: This paper focused only on cellular phenotypes in early infancy. Although we required children to have an 18 month visit for selection (to enable us to look at long-term clinical outcomes) we do not have 18mo flow cytometry data available.

3. EED/intestinal injury biomarker studies: What timepoints were examined? There could be differences between early life (breastfeeding) and post-weaning when food insecurity impedes nutrition, growth, exacerbates inflammation.

Response: As per the other immunology findings, the EED biomarker studies were from early infancy (30-89 days of age). Infants included in this study were universally breastfed, and had an optimal breastfeeding promotion intervention from community health workers which discouraged introduction of weaning foods before 6 months of age; almost 90% were reported to be *exclusively* breastfeeding at 1-3 months of age when these samples were collected (as published from this cohort previously, Mbuya et al. 2019). We have also added the following line to the Discussions section.

“Infants were universally breastfed, and samples for the immunological/intestinal analyses were collected before weaning took place, meaning they were not biased by differences in feeding or weaning between groups.” Page 18, lines 32-34.

4. Supplemental Figure 16: unclear how the last two sets of columns (both labelled “Female: Adjusted Hazard ratio”) differ

Response: Thank you for pointing out this typo, which has now been rectified (one column should have been male, not female).

5. There is no mention of accounting for multiple comparisons- with the large number of comparisons for the phenotype data it would be good to do some type of adjustment, maybe Holm-Bonferroni?

Response: We thank the reviewer for this comment. We acknowledge that the scope of this manuscript means that there are multiple comparisons, particularly for immune cell phenotype data in this paper, but also that, as highlighted by Rothman, “reducing the type I error for null associations increases the type II error for those associations that are not null”. Overall, the goal of the study was to infer patterns of responses based on effect sizes, clinical relevance, and trends across groups of biomarkers

rather than focus conclusions on single standalone findings below a 'significant' P value. We therefore include all test statistics and P-values to allow the reader to fully interpret significance levels in context. However, to account for multiple comparisons of flow cytometry data highlighted by the Reviewer, we have now also used the Benjamini-Yekutieli procedure (with a false discovery rate of 0.25) to adjust the level at which we should highlight findings as statistically significant, which estimated a threshold of $P=0.009$ or below. We now report phenotypic differences only when the P-value is <0.009 and have adjusted the methods results and discussion in line with this. For example:

“There were minimal differences in monocyte subsets between HEU and HIV-unexposed infants; compared to monocyte populations in HIV-unexposed infants, the classical monocyte population in HEU infants was 4.4 percentage points lower (95%CI 0.2, 8.7), although the evidence for this was weak; $P=0.04...$ ”. Page 9, lines 7-11.

6. Page 23 line 11 type: first “death” in this sentence should be “age”

Response: This has been corrected.

7. Since maternal HIV indicators were not associated with increased CRP, it would be interesting to investigate available maternal nutritional indicators (for example: weight gain in pregnancy, % body fat, food insecurity etc) to see if maternal malnutrition could be related to increased inflammation in pregnancy.

Response: We agree it will be important to explore this, however, we do not have data on maternal weight gain or body composition. We are planning a separate study to explore drivers of inflammation in pregnancy using available data on other aspects of maternal health, however these analyses are outside the scope of the current manuscript.

8. Was early infant CMV acquisition associated with mortality?

Response: We were not able to explore this; there were too few deaths (N=13) after 3 months of age, when we tested for CMV acquisition, to be able to test this hypothesis.

Reviewer #2 (Remarks to the Author):

The manuscript by Evans et al. reports novel findings on the role of the inflammation and CMV viremia during pregnancy in the mortality and immune development in HIV-exposed infants. The manuscript provides interesting insights into the mechanism by which CMV might affect HIV survival trajectory and highlights immunological differences in infant following in utero HIV exposure. This is a single country (Zimbabwe) study, and it nestles within a convenience cohort from other projects. The manuscript is well written and provides wealth of valuable data, however, some of the discussion points and conclusions, especially, regarding

global policies on the management of pregnant women are premature and overstate the significance of these findings. The analyses in this study were not the primary aims of the original trial and could represent significant bias from the mothers and infants with available samples and data. Prospective clinical trials are needed to validate the results from this study and to explore the public health implications. Here are a few specific suggestions for the authors:

Abstract: Suggest specifying the inflammation in the following statement “We demonstrate independent associations between maternal HIV, inflammation and cytomegalovirus co-infection and infant immune development and mortality” – please, specify whether you associate observed inflammation with HIV. Given the lack of association of CRP elevation with HIV viremia and ART, lack of association of CRP with infant mortality in non-exposed infants and known association of CRP elevation with bacterial and fungal infections, it is worth to clarify the speculated nature of the inflammation among women with HIV.

Response

We have changed ‘inflammation’ to ‘elevated CRP’ in the abstract to be clear about the inflammatory marker tested. We do not have space in the Abstract to speculate on the cause of inflammation (for which we do not present data) but we have added a discussion in the text:

“The drivers of maternal inflammation remain uncertain but could include microbial and vascular causes; there is a known association between CRP and bacterial and fungal infections, which are more common in HIV infection, and there remains a complex interplay between malnutrition, infection, inflammation, and impaired placental blood flow...”. Page 16, lines 22-26.

Methods: Please, specify whether the information was available/collected on maternal comorbidities beyond CMV and nutritional status. Please, clarify whether infants in the study were breastfed.

Response: We have added information (page 19, lines 22-23) on breastfeeding, which was universal, with a high prevalence of exclusive breastfeeding (Mbuya et al. 2019), due to promotion by the trial. We do not have information on other maternal comorbidities or infections, beyond what is presented in Table 1.

Results:

P 6 – lines 26-27 - Please, consider changing “We therefore questioned whether maternal inflammation during pregnancy is associated with infant mortality” to the “We hypothesized that”

Response: This has been changed as suggested.

P 10 – line 12 – how were the infants selected for the sampling of “117 HEU infants and 102 HIV-unexposed infants”?

Response

All HEU infant samples were included if children were confirmed HIV-negative at 18 months of age, had length measured at 18 months of age, had laboratory samples available from the 1-month postpartum visit, and had mothers with available laboratory samples from the trial baseline visit. Where possible, HIV-unexposed samples were selected as a comparison based on matching criteria. The main reason for not being included was the unavailability of a sample from 1 month of age.

The following appears in the Methods section of the revised manuscript:

“For immunological analyses, all infants in the SHINE trial born to women living with HIV were included if they were confirmed HIV-negative at 18 months of age, had length measured at 18 months of age, had laboratory samples available from the 1-month postpartum visit, and had mothers with available laboratory samples from the trial baseline visit. From the pool of infants born to women without HIV, infants were eligible for inclusion if they had length measured at 18 months of age, had laboratory samples available from the 1-month visit, and had mothers with available laboratory samples from the trial baseline visit. One infant was chosen for each HIV-exposed uninfected infant, matched on trial arm, sex, and season of birth. If more than one HIV-unexposed infant met the matching criteria, one of the infants was chosen at random. Infants who died or were HIV-infected were excluded.” Page 19 line 29-page 20 line 2).

Please, consider adding a figure with the study flow chart with the cohorts of infants and women.

Response: Thank you for this suggestion. We have added this as Supplementary Figure 1.

In the study only 18% of women with HIV had significant CMV viremia. The numbers of CMV studies are low in relationship to infant markers – “Among HIV-positive women with available infant immunophenotyping data, we identified CMV viremia in 32/102 (31%) women in the second trimester, and 20/85 (24%) in the third trimester”.

Response

The reviewer is correct that most women with detectable CMV had very low levels of viraemia. We believe this makes these finding particularly interesting, since we see associations with immune development and mortality even with very low-level viraemia, rather than *only* in association with significant levels of CMV viraemia, which are uncommon.

Discussion:

On page 16, the authors call for the public health approach measuring CRP during pregnancy in women living with HIV. How will the detected viremia be addressed in resource-limited settings? And do authors propose to stratify infant care based on single or repeat maternal measurement of CRP during pregnancy? In the Discussion authors are already

labelling the pregnancies with elevated CRP as “high-risk” based on this single study. The authors also propose antenatal interventions beyond ART, but do not specify which ones.

Response: We accept the Reviewer’s point that the public health strategy cannot be determined from this single study. Screening tests have their own criteria which must be assessed before implementation; however, we believe that it is intriguing that the CRP association with mortality was independent of viraemia, suggesting a separate pathway to mortality. However, the most important aspect of care remains early HIV diagnosis, ART initiation and virological suppression and we have stated this in the revised Discussion. We have rephrased this whole section and called for further studies to evaluate the utility of CRP as an additional measure of infant risk stratification (page 16, lines 18-21).

In the Discussion the HIV cure topic is not addressed, yet studies are looking at using CNV co-infection with HIV as a solution to immune therapy of HIV. Please, consider addressing.

Response: This is an interesting point, although we feel this might be a complex new issue to introduce since our paper does not focus on cure. Due to the limited word count for the manuscript, we have decided not to add this extra point about CMV (though we agree it is interesting).

The authors also speculate in regards of identified gender differences as following “exploring these differences may be essential in further designing interventions for this population”. I am unaware of any neonatal interventions that are gender/sex based and find this approach unethical.

Response

We accept this point, which was poorly expressed; we meant that the differential response between boys and girls might help us to understand intervention selection better. We have rephrased this to read:

“Exploring the differential responses between boys and girls may help to identify underlying disease mechanisms, which would inform future intervention strategies”. Page 18, lines 27-28.

Reviewer #3 (Remarks to the Author):

This is an important and significant paper in the field of CMV and HEU children and immune development. The authors explore the impact of maternal immune activation and CMV infection on the shape of infant immunity between HEU and HUU controls in Zimbabwe, using participants from the SHINE trial. The noteworthy results are that maternal CRP from HIV infected mothers is associated with infant mortality; HEU infant have activated and more mature CD8+ T cells along with activated monocyte populations; significant differences between boys and girls; CMV viral detection during gestation is independently associated

with mortality; CMV exposure during pregnancy and acquisition in the first 3 months of life associate with CD8+ T cell activation and altered phenotypes. The MS was however, very difficult and annoying to read as the majority of tables did not correspond to the narrative results. This warrants a major revision.

Major Comments

1. Adverse birth outcomes, such as pre-term birth, are not factored into the analysis as a major driver in infant mortality. Is CMV detection in 2nd or 3rd trimester associated with PTB and poor infant survival? If this is unknown, there needs to be a mention of this in the discussion.

Response

There were no associations between CMV in the second trimester and preterm birth. There was a very weak association between CMV in the third trimester and preterm birth (OR 1.01, 95%CI 1.00, 1.09), and a strong association between prematurity and mortality (OR 4.83, 9%CI 2.67, 8.74). Because pregnant women in our study did not have access to ultrasound scanning, our data on gestational ages are not completely reliable, and may be capturing an element of small-for-gestational age also. Nevertheless, we re-ran models including preterm birth, and the estimates are largely unchanged from the original manuscript. For example, the association between third trimester CMV exposure and mortality after adjusting for prematurity was aHR 2.11 (95%CI 1.10, 4.02) compared to 2.11 (95%CI 1.09, 4.09) without adjustment. However, we did not add birth outcomes into models as a *confounder* because we believe they may be on the causal pathway between CMV exposure and infant mortality, and inclusion would therefore over-adjust our models; more studies are required to identify these mechanistic pathways. We have added a line regarding this in the Discussion section.

“There are several other plausible explanations, and these are likely to be multifactorial, including an impact of CMV exposure on birth outcomes such as low birthweight, small-for-gestational age, and prematurity.” Page 17, lines 32-34.

2. Is the type of ARV regimen impactful on maternal and HEU immune activation? Those on a TFV-based regimen are 397 and those on an AZT-based regimen are 120, so an ability to identify maternal ARV regimen as a driver or a confounder would be good to build in.

Thank you for this question, which we agree is interesting. We have now re-run several analyses to explore this as a potential driver and/or confounder, but found no meaningful differences. When looking at mortality, there was no difference between the two groups (21 deaths from 397 children exposed to TDF-based regimens (5%) and 6 deaths from 120 children exposed to an AZT-based regimen (5%). Because we included HIV viral load in models, ART was closely linked and so did not have an impact in the models when used as a potential confounder. Similarly for the immunology comparisons, we did not identify meaningful differences in measures of immune activation according to ART regimen. Since our manuscript is already lengthy, we have decided not to add this extra point about ART regimens.

3. There is almost an endless list of tables (23 in total) and the authors need to prioritize which of these are central to their story and which are supplementary and which can be omitted entirely (or amalgamated). Table 24 seems to be a repeat of table 1 and table 25 is mislabeled. As mentioned above, very few tables correspond with the description in the results.

Response: Thank you for this comment. We have reduced the number of tables to 13. We have also removed the accidental duplication and rectified the mislabelled tables.

4. The title of the MS is based on inflammation and CMV, but CMV doesn't come into the picture until halfway through the results, where there is a long list of how maternal HIV impacts, or does not impact, on immune phenotypes in the HEU. The authors may wish to trim this down or consolidate the data.

Response: Thank you for this comment. We tried re-writing the manuscript in several ways, but we felt that the current way is the clearest way of reporting our findings. For a paper of this scope, it is not possible to spell out all concepts at once, and so our rationale was to build the story by looking at HIV first, followed by inflammation and immunology, followed by CMV; even though the earliest statements of the findings are not as novel as later findings, we felt it was important to show these and confirm that those well-recognised risk factors are apparent in this contemporary cohort, before going into the more novel findings regarding CMV. We would prefer to retain this foundational data but are open to reviewer/editorial guidance if they feel strongly that it should be changed.

Minor comments

1. Some of the compensation settings in Figure 4 look suspect – such as CD4 vs CD8, where CD8 is spilling into the CD4 channel. CD57 looks over-compensated. What was the QC/QA on the flow assays and were compensation beads used?

Response

Thank you for this comment. We agree that the sample chosen to demonstrate our gating strategy is demonstrative of our method but was not selected as a 'perfect' example. To clarify our QC/QA for compensation across all buffy coat cell samples, we confirmed consistent cytometer performance and voltage settings across each batch of samples by running a Performance QC program prior to sample analysis each day; samples were only run when this program passed. To check for spectral overlap between fluorescence channels we ran unlabelled and single-fluorochrome-labelled compensation beads (UltraComp beads, eBiosciences, Cat# 01-2222-42) for each fluorescence channel (FITC, PE, PerCP-Cy5.5, PE-Cy7, APC, APC-Cy7; 10,000 events acquired per compensation control) prior to each batch of labelled samples. Positioning of the unlabelled/labelled compensation beads was checked manually for correct position on relevant fluorescence axes during the flow cytometer run, confirming correct PMT voltages for each detection channel. We confirmed consistency of compensation across sample batches by generating automated

compensation matrices for each batch of compensation beads in FlowJo software; these were confirmed to be essentially identical. We therefore applied the same validated compensation matrix to raw (uncompensated) .fcs files generated for all samples in FlowJo software. We used the same cytometer and used consistent voltages on each run. Given the size of our cohort, recognised technical variation in processing human blood cells, and biological variation inherent to a community cohort like ours, we recognise that compensation may not be perfectly applied across all samples which is why our study was powered for analysis of differences and associations on this background heterogeneity.

For completeness on our QC/QA, our method for confirming our flow cytometry gating was as follows: after running performance QC for the flow cytometer and compensation beads, we ran 14 FMO controls (pooled buffy coat cells from all participants in the batch labelled with each of the 4 antibody cocktails for monocyte and T cell immunophenotyping) and 1 unlabelled sample (pooled cells). Unlabelled cells and FMO controls were used to determine the positions of gates for immunophenotyping panels.

2. There is also no viability stain (ie. being able to discriminate live/dead). This is particularly concerning as spurious staining may result from antibodies non-specifically binding to dead cells. How did the authors account for this

Response

As outlined in our Methods, flow cytometry analysis was carried out on fixed cryopreserved buffy coat cells (i.e. 0% viability at time of flow cytometry analysis); upon arrival in the laboratory in anticoagulant treated blood collection tubes, whole blood was treated to simultaneously lyse red blood cells and fix leukocytes (BD FACS Lysis buffer including PFA as a fixative) and then cryopreserved in freezing medium (containing DMSO and FCS to maintain structural integrity of fixed cells).

Whilst fixation and cryopreservation have a recognised impact on surface marker expression, these steps were required given our field setting (i.e. no on-site flow cytometer) and collection of longitudinal samples over several years. Our cryopreservation methods were optimised based on a field study in Ugandan infants which had similar laboratory capacity to our field hub laboratory in rural Zimbabwe (Lutwama F et al., J Infect Dis 2014), and technical discussion with the study authors. Our staining protocol included typical steps to avoid non-specific antibody binding (i.e. a blocking step in PBS containing FCS; FCS included in all flow cytometry reagents) and gating was validated relative to FMO controls for non-specific binding. Other groups using a similar protocol (Nemes et al. 2015 Cytometry A; 87(2):157-65) have reported that cell subsets can be “reliably quantified and characterized in fixed whole blood by intracellular staining and flow cytometry, even after sample cryopreservation...facilitating advanced immunophenotyping in clinical and research settings with limited technical resources...”. Further, we optimised each immunophenotyping panel prior to implementation in our study by comparing antibody labelling of fresh and cryopreserved buffy coat cell samples; where antibody

labelling underperformed on cryopreserved versus fresh samples, we titrated antibodies to increase sensitivity for the affected marker and/or trialled an alternative clone. Where a cell marker continued to be undetectable in cryopreserved samples (e.g. some cell surface markers are more susceptible to cleavage by fixative reagents than others), we excluded that marker and re-configured the relevant antibody panel accordingly; for example, this is why we chose to use CD27 instead of CCR7 which is more typically used in T cell differentiation immunophenotyping panels for live/non-fixed cells.

Collectively, we are confident that our results do not reflect spurious binding of antibodies to dead cells.

3. What are the changes in the numbers of cells, only frequencies are given

Response: Thank you for this comment. We standardised sample processing based on blood volume as we did not have access to a cell counter, we therefore did not collect data on white cell counts or lymphocyte counts and are only able to report proportions.

4. Not sure sex differences is a supplementary table

Response: We have now moved several of sex differences findings into main tables, although we could not move them all due to the limit on numbers of tables/figures.

5. Lines 25-27 on page 15 – this is a very tenuous connection (as correlations do not suggest cause) and should appear in the discussion and not the results

Response: We have removed this from the Results section.

6. The authors could use an unbiased approach to the phenotype analysis, rather than manual gating.

Response: Thank you for this suggestion. We recognise that there are now machine-learning/AI-based approaches to automate flow cytometry gating however we did not have capacity for this form of analysis at the time we analysed these samples, which was undertaken in Zimbabwe. However, we did take care throughout our analyses to avoid bias, as follows. (1) All samples named according to de-identified participant code (PID) meaning that flow cytometrists were blinded to, age, sex, HIV, CMV statuses, etc. during sample processing; (2) all .fcs files were named according to de-identified participant code (PID) meaning that gating in FlowJo also took place whilst blinded in the same way; (3) manual gating of each sample was standardised relative to FMO controls; (4) statistical analyses of samples according to participant clinical and demographic characteristics was only conducted after immunophenotyping data had been exported from blinded analysis in FlowJo.

REVIEWERS' COMMENTS

Reviewer #3 (Remarks to the Author):

I am happy with the responses and rationale used in the rebuttal and the MS reads very well and is an important contribution to the field.

The only point I would make for future flow analysis from fixed whole blood cells is to add the live/dead cell dye before you fix and then you would be sure to gate out non-specific staining (even if 1%). It means losing one FL channel, however.